# The photolyase/cryptochrome of *Aspergillus nidulans* senses oxidative stress and shuttles from nuclei to mitochondria

Alexander Landmark [1], Tim Rudolf[1], Kevin Hundshammer[1], Jasmin Böhm[2], Kai Leister[1], Sylvia Erhardt [2] & Reinhard Fischer [1] ✉

Cryptochromes are photoreceptors with functions in the entrainment of circadian clocks or as proposed magnetoreceptors in birds or as light-independent regulators of stress responses in plants. Here, we show that the fungal cryptochrome-like photolyase CryA from *Aspergillus nidulans* is induced by light and oxidative stress and establishes negative-feedback loops for light- and stress-activated genes. The negative-feedback loops depend on CryA interaction with phytochrome and the HOG (high osmolarity glycerol) pathway transcription factor AtfA in nuclei. CryA translocated in less than one minute from nuclei to mitochondria in the presence of hydrogen peroxide suggesting mitochondrial functions and possibly mitochondrial-nuclear communication. The shuttle to mitochondria depended on the N-terminal extension and a cysteine therein, which probably induces conformational changes of CryA upon oxidation. Therefore, we propose CryA as a sensor for oxidative stress. Such an N-terminal extension is also present in other photolyases and some cryptochromes, suggesting evolutionary conservation of the mechanism.

For sessile organisms, such as plants or fungi, the perception of sunlight is crucial, and adaptation to light and the associated environmental changes is important for growth and survival. To accomplish this task, these organisms use specialized photoreceptors that have evolved in all major kingdoms of life[1]. Such receptors contain an organic, light-absorbing chromophore that responds to specific wavelengths, inducing conformational changes and ultimately activation of the protein[2–4]. One such group is the blue light-sensing cryptochrome and photolyase family (CPF), a highly conserved group of light-driven proteins found in all clades of life[5,6]. They are classified as flavoproteins and share a conserved photolyase homology region (PHR) that usually binds the main chromophore flavin (FAD) and a secondary antenna chromophore 5,10-methenyltetrahydrofolate (MTHF)[7]. In contrast to photolyases, cryptochromes have an additional C-terminal extension to interact with regulatory proteins[5,8]. Photolyases repair either cyclobutane pyrimidine dimers (CPD) or 6–4 photoproducts after UV light exposure by using the light energy to cleave the damaged DNA bases[9]. The role of animal

cryptochromes, particularly in the regulation of the circadian clock, has been extensively characterized, but they have additional functions such as the perception of the earth magnetic field in insects or birds, or as light-independent transcriptional regulators for cellular responses[10–14]. Plant cryptochromes, together with the red-light-sensing phytochromes, entrain the circadian clock to control multiple physiological functions such as the secondary metabolism for pathogen resistance, promotion of flowering, seed dormancy, and the osmotic stress response[6,11]. While plant and animal photoreceptors and their functions have been thoroughly described and analyzed, there are only few studies in fungi. Only recent advances in genome-wide analyses and sequencing have provided new insights into fungal light responses and their signaling pathways[1,15]. Light signaling is involved in many aspects of fungal development, including sexual and asexual differentiation, circadian rhythm, secondary metabolism, and pathogenicity[16–18].

Fungal light receptors are divided into three major groups: the red/far-red light-sensitive phytochromes with a linear tetrapyrrole,

[1]Department of Microbiology, Karlsruhe Institute of Technology (KIT) - South Campus, Karlsruhe, Germany. [2]Karlsruhe Institute of Technology (KIT), Zoological Institute and Institute of Biological and Chemical Systems-Functional Molecular Systems, Karlsruhe, Germany. ✉e-mail: Reinhard.fischer@kit.edu

probably bilin, as chromophore; the blue light-sensing LOV proteins, the cryptochrome/photolyase family, and the less common BLUF proteins, which all utilize a flavin molecule for light perception; and the retinal-binding, green light-sensing proton pumps called opsins[1,19,20]. In particular, the function of the light, oxygen, and voltage (LOV) domain-containing white-collar complex (WCC) and the phytochrome response together with the HOG pathway have been investigated in recent years. The blue-light receptor white collar 1 (WC-1) is best known as a component of the circadian clock of *Neurospora crassa* or *Botrytis cinerea*[21–24]. In comparison, phytochrome is activated by red light or temperature, resulting in the induction of temperature and/or light-dependent genes[25–27]. Knowledge about fungal cryptochromes or photolyases and their physiological roles is rather limited[1,15]. Fungi usually encode a *Drosophila*, *Arabidopsis*, *Synechocystis*, *Homo* (cry-DASH), and a CPD- or 6–4 photolyase for UV-light induced DNA damage[5]. *B. cinerea* harbors two CPF proteins, of which CRY1 is responsible for DNA repair, and CRY2 acts as a repressor of photo-induced conidiation and growth[28]. CryA from *Arthrobotrys oligospora* and CryD from *Fusarium fujikuroi* control the production of secondary metabolites and act as negative regulators of conidiation or pathogenicity[29,30]. The CRY-DASH protein CRY-1 from *N. crassa*, together with the green-light receptor NOP-1 and the red-light receptor PHY-2, inhibits the WCC-controlled circadian rhythm, although the precise mechanism is not fully understood[31,32]. Cryptochromes from *Mucor circinelloides* and *Phycomyces blakesleeanus* play a role in photoreactivation and bind double-stranded DNA[33–35].

Surprisingly, there are several cases of fungal photolyases with additional regulatory functions. Such a dual role of CPF members was previously only observed in algae and animals, indicating the evolutionary ancestry between photolyases and cryptochromes[36–38]. In *Trichoderma atroviride*, the 6–4 photolyase Cry1 and the CPD photolyase Phr1 are responsible for photoreactivation and have additional blue- and red-light dependent regulatory functions[39–41]. PHL1 from *Cercospora zeae-maydis* responds to UV irradiation by inducing the expression of repair genes[42]. CryA from *A. nidulans* was described as a canonical CPD I photolyase, but also negatively modulated sexual development[43]. Taken together, CPFs in fungi have multiple functions, but only in a few cases their molecular roles were described.

*A. nidulans* is a filamentous fungus and a model organism for light signaling in fungi[44–46]. *A. nidulans* has a pronounced red-light response, and fungal phytochrome A (FphA) is the main regulator[45]. FphA acts in a histidine-aspartic acid-dependent phosphotransfer-relay system that triggers the stress-induced HOG pathway. This pathway transduces the light signal into the nucleus via the MAPK cascade to activate the ATF/CREB transcription factor AtfA, resulting in increased resistance to environmental stress and inhibition of sexual development[26,45,47,48]. Phytochrome can also shuttle into the nucleus to regulate gene transcription by chromatin remodeling[49,50]. These responses should be transient and therefore need to be balanced or stopped. In *N. crassa*, the blue-light receptor Vivid (VVD) and the clock protein Frequency (FRQ) act as inhibitors of the WCC in a negative-feedback loop[51,52]. This raises the question of whether an analogous mechanism is present for phytochrome-dependent gene regulation in *A. nidulans*, where VVD and FRQ are absent. One possibility for a blue-light-driven negative element is the cryptochrome/photolyase family. *A. nidulans* harbors only one CPF member, the CPD I photolyase CryA, in addition to the two WCC orthologues. This is largely unique for the *Aspergillus* family, as other organisms usually encode two or more cryptochrome or photolyase-like genes[43,44]. Although classified as a DNA repair enzyme, CryA was described as the first dual-function photolyase/cryptochrome that regulates sexual and asexual differentiation by inhibiting the sexual transcription factor Velvet A (VeA). Deletion of *cryA* results in an increased number of cleistothecia[43]. Recent studies have shown that *cryA* expression responds directly to oxidative stress and light through the activation of AtfA[15,53].

Here we show that CryA is involved in a negative-feedback loop to regulate light and stress responses of *A. nidulans*. The expression of *cryA* is light-stimulated, and CryA is a nuclear protein. Deletion and overexpression of *cryA* interfere with the development of *A. nidulans*. CryA directly interacts with FphA and AtfA in the nucleus and negatively regulates the light response, likely through histone modification by FphA. Interestingly, CryA also responds to reactive oxygen species (ROS) and negatively affects resistance to oxidative stress by interacting with AtfA. Upon $H_2O_2$ application, CryA leaves nuclei and rapidly travels to mitochondria. This behavior depends on a cysteine residue in an N-terminal extension of the protein. This characterizes CryA as a main regulator for the general stress and light response of *A. nidulans* and unravels a nuclear-mitochondrial shuttle as a novel regulatory principle and CryA as a putative ROS sensor.

## Results

### *A. nidulans* CryA is a CPD photolyase with FAD and MTHF as chromophores

Phylogenetic analyses of cryptochrome/photolyase members grouped *A. nidulans* CryA into class I CPD photolyases[43]. To further identify similarities and differences in the domain architecture of CryA, we used InterPro and SUPERFAMILY to compare CryA from *A. nidulans* with CPF proteins from *Arabidopsis thaliana*, *Homo sapiens*, and *Escherichia coli*, Phr1 from *T. atroviride*, and PHR1 and CRY2 from *Aspergillus niger* (Fig. 1a). All CPF members consist of a highly conserved N-terminal photolyase domain (light pink) and a C-terminally located FAD-binding domain (blue). In addition, cryptochromes contain a non-specific and variable C-terminal extension (CTE), which is absent in photolyases and in CRY-DASH proteins. CryA harbors an N-terminal extension (NTE) that is missing in prokaryotic but is present in all fungal CPD photolyases and in some CRY-DASH proteins[54].

We then used AlphaFold 3 and AlphaFill to model the structure of CryA and predict the potential chromophores (Fig. 1b and Supplementary Fig. 1a). The core protein structure is highly conserved (pLDDT >0.95) with lower confidence for the N- and C-terminal tail of CryA. AlphaFill predicts FAD (orange), FMN (green), and 5,10-MTHF (purple) as major co-factors, with FAD and MTHF having a higher confidence, defined by lower global and local RMSd and TCS values compared to FMN. These predictions are consistent with the experimental data from other fungal CPF members and photolyases in general, identifying CryA as a folate photolyase with FAD as the main co-factor and 5,10-MTHF as the antenna chromophore[33,34].

Since *Aspergilli* often only have one CPD I photolyase, and some members have a second photolyase/cryptochrome-like gene encoding a CRY-DASH cryptochrome, we analyzed the evolutionary relationship between CryA and other CPFs of the *Aspergillus* family. In addition to prominent members of animal, plant, and bacterial CPFs, we primarily selected representatives of the fungal kingdom that have been previously described for their regulatory and/or repair activity[5,28,29,42,55]. We used the bioinformatic tool Geneious Prime to generate a phylogenetic tree comparing CPD photolyases (green) with animal cryptochromes (purple), plant cryptochromes (red), and CRY-DASHs (blue) (Fig. 1c and Supplementary Table 1). All predicted photolyases from the *Aspergillus* family cluster together with members of the CPD photolyase subfamily, supporting data from previous studies, describing CryA as a CPD I photolyase. Strikingly, the additional cryptochrome-like gene, which is found in *A. clavatus* and *A. niger*, is located in the animal cryptochrome subfamily. This suggests an evolutionary divergence between the *Aspergillus* progenitors and other fungal representatives, which primarily contain a CRY-DASH protein such as CRY from *N. crassa* or CryA from *M. circinelloides*[31,33].

Next, we aimed at validating the predicted properties of CryA experimentally and purified *A. nidulans* CryA tagged with six histidines from *E. coli*. CryA protein had an apparent molecular mass of 60-65 kDa, which is in good agreement with the calculated molecular mass of

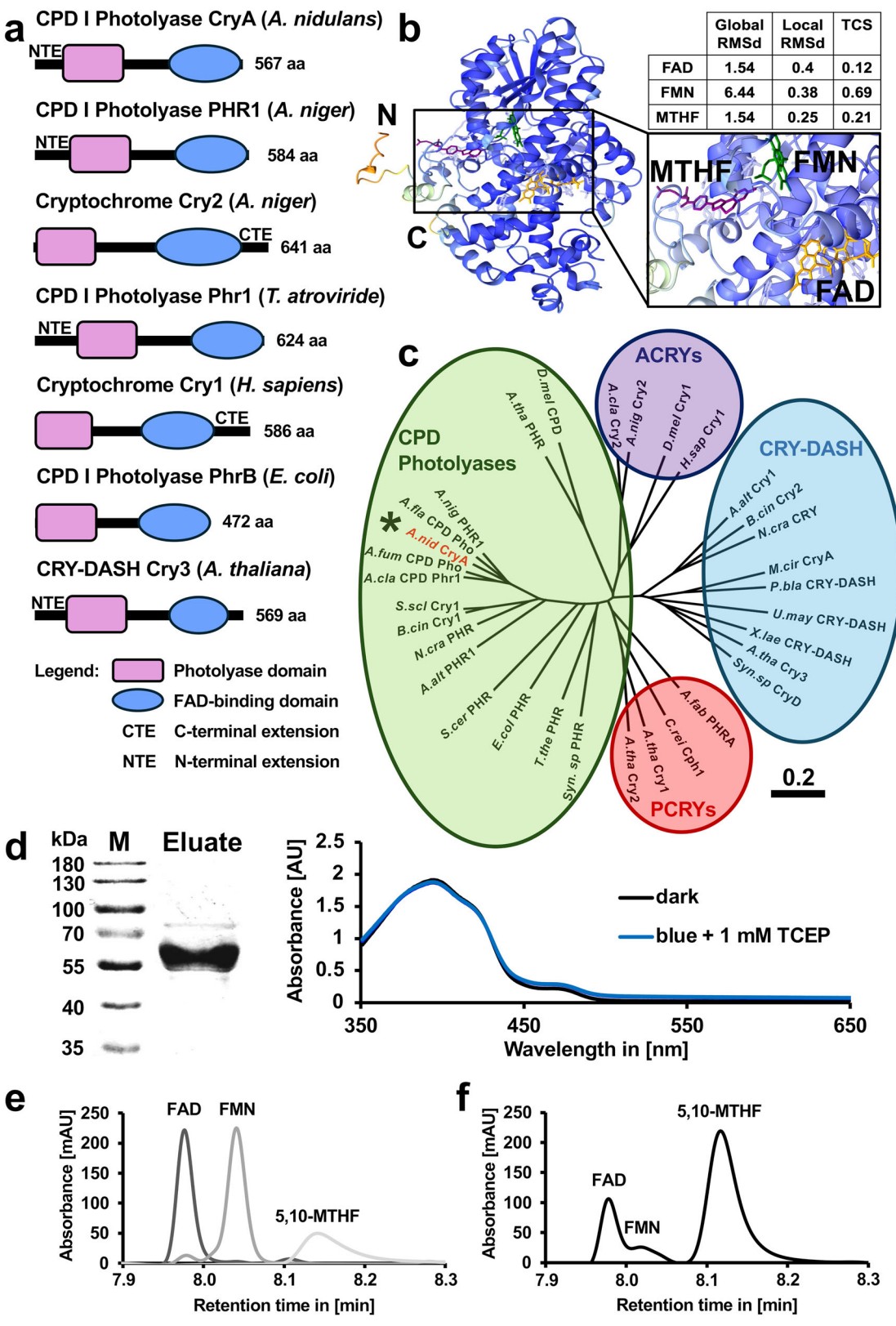

65 kDa (Fig. 1d). CPD photolyases usually use FAD as the main chromophore, transferring the energy of reduced FADH$^*$ to CPD lesions to repair DNA damage. The absorption spectrum of CryA showed a protein-bound FADox typical peak at around 470 nm and a significantly stronger peak at 385 nm, indicative of the antenna pigment MTHF[33,56]. To experimentally validate the co-factors, we separated the chromophores from the photoreceptor and compared them to FAD and FMN using

emission- and absorption spectroscopy (Supplementary Fig. 1b). Flavin molecules have a similar spectral pattern which can be distinguished by the shift in absorption at different pH[57–59]. The isolated co-factors showed a comparable pH-dependent increase of fluorescence like the FAD control. The released chromophores had a characteristic absorption spectrum similar to the one of FAD or FMN with a slightly stronger absorption at 365 nm, indicating the antenna chromophore 5,10-MTHF.

**Fig. 1 | CryA is a canonical CPD I photolyase. a** InterPro and SUPERFAMILY were used to analyze and compare domain structures of CryA from *A. nidulans* with CPF members from, *T. atroviride* (Phr1), *A. niger* (Cry2 and PHR1), *H. sapiens* (Cry1), *E. coli* (PhrB), and *A. thaliana* (Cry3). The conserved elements are the photolyase domain (light pink) and the FAD-binding domain (blue). **b** AlphaFold 3 and AlphaFill prediction of CryA and possible chromophores. Coloration of the protein structure is based on confidence levels of AlphaFold measured through pLDDT (predicted local distance difference test). Blue indicates high confidence, and orange indicates low confidence. The quality of the predicted chromophores is represented by the Global and Local RMSd (root-mean-square deviation) and TCS (transparent clash score) value. Lower scores indicate higher confidence. FAD is colored in yellow, FMN in green, and MTHF in pink. **c** Geneious Prime 2024 was used to perform a phylogenetic analysis of members of the CPF with a special focus on fungal representatives. Global alignment was used for sequence alignment with a Pam-matrix of 100 and gap costs of 10/1. The phylogenetic tree was generated with Jukes-Cantor as a distance model and Neighbor-Joining as an algorithm. Subfamilies are CPD photolyases (green), pCRYs (red), aCRYs (violet), and CRY-DASH (blue). The scale bar represents substitutions per site. **d** SDS–PAGE on the left and the absorption spectrum on the right confirm the successful heterologous expression of CryA in *E. coli*. The absorption spectrum was measured on the spectrophotometer under red safety light. The experiment was repeated at least 10 times with similar results. **e** Identification of the chromophores of CryA using HPLC. Monoglutamated 5,10-methenyl tetrahydrofolate (5,10-MTHF) was prepared as a standard in 100% methanol with a concentration of 0.5 mM, flavine-adenine-dinucleotide disodium salt (FAD) and riboflavin 5′-monophosphate sodium salt hydrate (FMN) were used as standards with a concentration of 1 mM in dH$_2$O. The absorbance was measured at 380 nm. **f** Co-factors of CryA (concentration of 40 μM) were dissolved in water and separated as described before. The absorbance was measured at 380 nm. Source data are provided as a Source Data file.

We performed absorption, excitation, and emission spectroscopy to confirm MTHF as the second chromophore (Supplementary Fig. 1c). Although the excitation spectrum was identical to standard 5,10-MTHF, the emission spectrum was not conclusive enough. Since it was not possible to induce a detectable photocycle to analyze the chromophore behavior, we performed liquid chromatography to confirm FAD as the major chromophore and to visualize any potential secondary chromophores (Fig. 1e). We replaced the buffer with dH$_2$O and denatured the protein to release the chromophores[60]. FAD, FMN, and 5,10-MTHF were used as standards. The retention times of the chromophores released from CryA were identical to the FAD standard, but the retention time for MTHF had a slight shift to the second major peak of CryA. This observation can be explained that 5,10-MTHF in *E. coli* contains more glutamate residues than standard 5,10-MTHF[33,61,62]. The third noticeable peak had a similar retention time as FMN, indicating that CryA either uses FMN as an additional antenna chromophore or some expression artifact in *E. coli* leading to FMN loaded into the FAD chromophore pocket. These observations confirm that CryA uses FAD as the main and 5,10-MTHF and possibly FMN as antenna chromophores.

## CryA is a negative feedback regulator of light- and development-induced genes

Deletion of *cryA* caused drastic changes in the ratio between asexual and sexual development in *A. nidulans*. The deletion strain produced more sexual fruiting bodies in light on solid media and large numbers of Hülle cells in liquid media[43]. If CryA has a direct regulatory function, we anticipated that overexpression of the gene should also affect development. To test this hypothesis, we first fused CryA N-terminally to GFP to visualize the protein in vivo (Fig. 2a) CryA exclusively localized to nuclei. This behavior did not change during asexual development under different light conditions. (Supplementary Fig. 2a). Next, we investigated the function of CryA as a light receptor in *A. nidulans*. Previous RNAseq results showed that the induction of *cryA* depends on phytochrome and should therefore be regulated by light[15]. The expression of *cryA* was indeed transcriptionally upregulated in wild-type after light exposure, although the upregulation ratio was not very high (Fig. 2b). Next, we fused the open reading frame of *cryA* to the inducible alcohol dehydrogenase promoter *alcA* to test whether overexpression of *cryA* results in phenotypic changes. The promoter is repressed in the presence of glucose, de-repressed with glycerol, and induced with threonine[63]. On solid minimal media containing glucose (MM-Glu) or threonine (MM-Threo) as the main carbon source, wild-type *A. nidulans*, the *cryA*-deletion mutant, the overexpression strain *cryA*OE, and a Δ*cryA* re-complemented strain ( + *cryA*) grown under different light conditions were compared (Fig. 2c). While there were no noticeable phenotypic differences for the WT and the *cryA* mutants on minimal media with glucose, *cryA* overexpression exhibited severe growth and developmental alterations on threonine media. This phenotypic change was irrespective of light. Wild-type *A. nidulans* displayed the typical green pigmentation on minimal glucose medium,

which changed slightly to a lighter pigmentation (fewer conidia) on threonine medium. Overexpression of *cryA* completely abolished the formation of conidiophores and conidia, resulting in a white appearance of the colonies. This phenotype was also visible under the microscope, with hyperbranched hyphae forming conidia-like structures at the tips. Deletion of *cryA* also showed reduced conidia formation, particularly on threonine medium, and Hülle cell formation in liquid culture. This phenotype was rescued with a wild-type copy of *cryA* (+*cryA* strain).

Since threonine is not optimal to investigate developmental changes in *A. nidulans*, we used complete media (CM) with either glucose or glycerol (CM-Gly) under low oxygen and varying light conditions (Fig. 2c and Supplementary Fig. 2b). The *cryA*OE strain behaved similarly on complete media with glycerol, with a complete abolishment of pigment and conidia formation. In addition, the Δ*cryA* strain produced much more cleistothecia than the wild-type. The effect was quantified in strains grown in the dark or under blue- or red-light conditions (Supplementary Fig. 2c). Deletion of *cryA* led to significantly higher numbers of cleistothecia irrespective of the growth conditions. This phenotype was partially rescued in the re-complementation strain. In contrast, overexpression of *cryA* resulted in no visible cleistothecia, further establishing the role of CryA in the development regulation of *A. nidulans*.

Next, we tested the expression of several light-responsive genes in the *cryA*-deletion mutant, the re-complemented, the *cryA*-overexpression, and the wild-type strain (Fig. 2d). We selected *ccgA*, *ccgB*, and *AN11314* as marker genes for short-term light activation, and *conJ*, *brlA*, and *flbB*, which are associated with light-induced development[15,64]. The wild-type showed significant upregulation of expression for most of the analyzed genes after illumination. This activation was significantly stronger in the Δ*cryA* mutant and significantly weaker for most genes in the *cryA*OE strain. Re-complementation of the *cryA* deletion resulted in an expression profile similar to wild-type. Because the deletion of *cryA* triggered a strong upregulation after short light treatment, we hypothesized that CryA could be involved in photoadaptation, the reduction of the expression during continuous light exposure (Supplementary Fig. 2d). Indeed, *ccgA* was upregulated after 15 min of illumination, and the transcript level decreased largely after 60 min. However, the Δ*cryA* strain showed a very similar behavior. In the case of the *brlA* gene, photoadaptation was not observed in wild-type, but the expression increased for 120 min. In this case, CryA appears to be involved in the long-term upregulation of *brlA* because the mRNA levels dropped to almost zero after 60 min. The *cryA*OE strain exhibited very low expression levels at all time points. Hence, CryA inhibits the expression of light-induced and development-induced genes but appears not to play a direct role in photoadaptation.

Altogether, altering the protein level of CryA results in genetic mis-regulation of light-induced genes and severe developmental changes. CryA accumulates after light exposure, limiting the

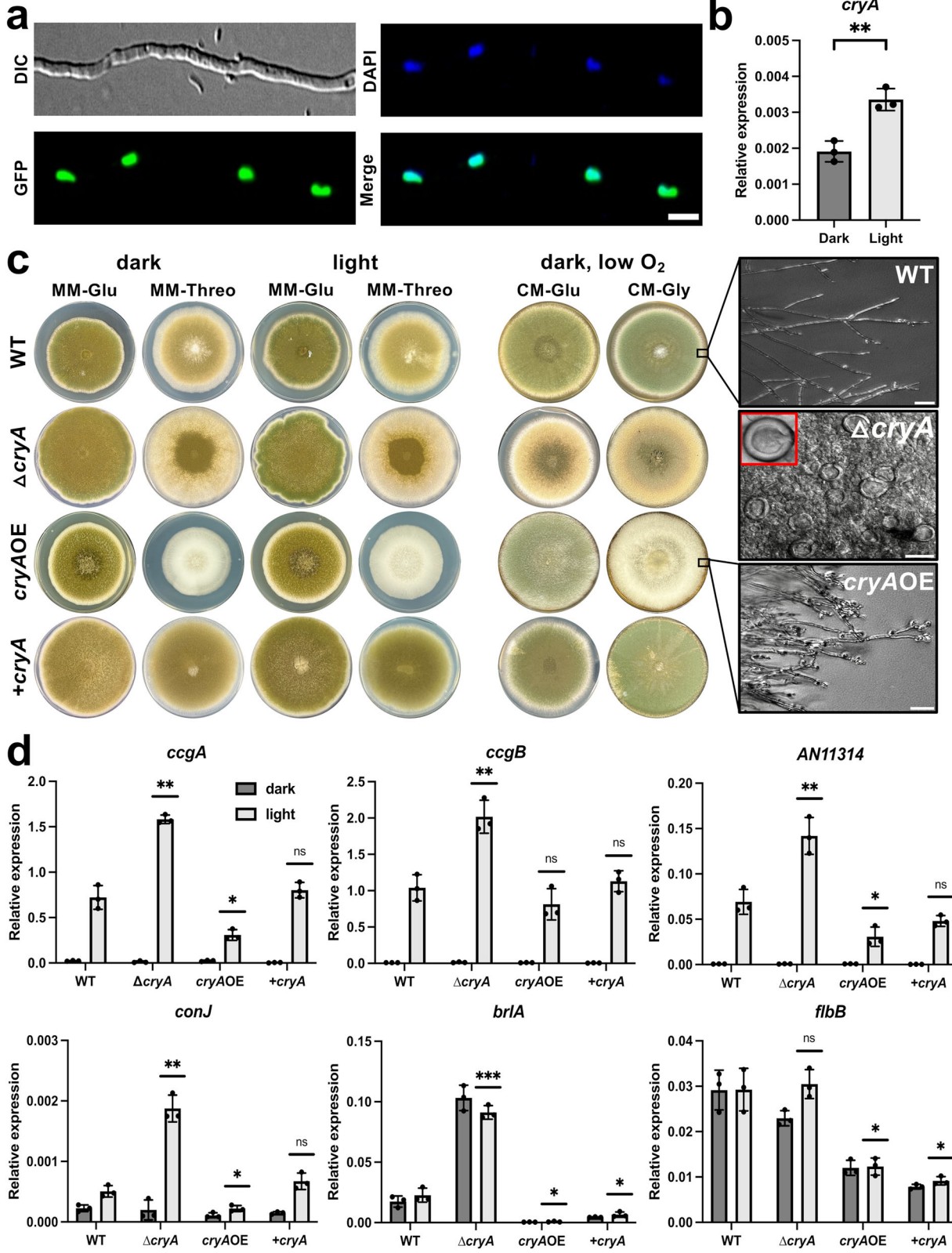

photoresponse and inhibiting development. Hence, the system resembles a typical negative-feedback loop.

### CryA interacts with phytochrome and negatively affects the red-light response

The negative effect of CryA on the light response could be due to either genetic or physical interaction of CryA and FphA. To distinguish

between the two possibilities, we used AlphaFold 3 to test the interaction of the phytochrome dimer with CryA (Fig. 3a and Supplementary Fig. 3a). For improved visualization, only the conserved PAS, GAF, and PHY (PGP) domains of FphA (195–753 aa) were used. AlphaFold predicted with low confidence an interaction with a predicted template modeling (pTM) score of 0.44 and an interface predicted template modeling (ipTM) score of 0.33, suggesting that CryA interacts

**Fig. 2 | CryA is a nuclear protein and is involved in developmental regulation and light sensing. a** CryA was fused N-terminally to GFP under the control of the *alcA* promoter. Spores were incubated overnight at 25 °C in minimal media with 2% glycerol instead of glucose to induce the promoter. Nuclei were stained with HOECHST. The scale bar represents 5 μm. The experiment was repeated at least ten times with similar results. **b** Analysis of the *cryA*-expression level in darkness and after light treatment. In total, $1.5 \times 10^8$ spores of WT strain were incubated for 16 h in 10 ml minimal media at 37 °C in the dark. Light samples were illuminated for 15 min with blue and red light (200 μmol photons/m²*s²) simultaneously before harvesting the mycelia under green safety light. For the qRT-PCR the *h2b* gene was used as a housekeeping gene for normalizing the expression. Error bars represent the standard deviation of three biological and two technical replicates. For statistical analysis, an unpaired, parametric two-tailed Student's *t* test was performed, using a 95% confidence interval, **$P \leq 0.01$. Dots represent each individual biological replicate. **c** Comparison of wild-type colonies with a *cryA*-deletion strain (Δ*cryA*), a *cryA* re-complementation strain ( + *cryA*), and a *cryA*-overexpression strain (*cryA*OE). In total, 1000 spores were grown for 7 days at 37 °C on solid media with either 2% glucose (MM-Glu) or 2% threonine (MM-Threo). For the analysis of cleistothecia formation, $10^5$ spores were inoculated on complete media with either 1% glucose (CM-Glu) or 1% glycerol (CM-Gly), and plates were enclosed with parafilm to exclude exogenous oxygen. Colonies grown on solid media (WT and *cryA*OE) or samples from liquid medium (Δ*cryA* strain, 3 days incubation in the dark) were analyzed microscopically. Microscopy was performed on an Axio Imager Z1 with the objective W Plan-Apochromat 10x/0.5 in the DIC channel. Scale bar represents 100 μm. **d** Investigation of light-regulated and conidiation-related genes in wild-type compared to a *cryA*-deletion strain, the re-complementation strain +*cryA*, and a *cryA*-overexpression strain. In all, $1.5 \times 10^8$ spores of the respective strains were incubated for 16 h in 10 ml minimal media with 2% threonine at 37 °C in the dark. Light-treated samples were illuminated for 15 min with blue- and red light (200 μmol photons/m²*s²) simultaneously before harvesting the mycelia under green safety light. For the qRT-PCR, *h2b* was used as a housekeeping gene for normalization. Error bars represent the standard deviation of three biological and two technical replicates. For statistical analysis, an unpaired, parametric two-tailed Student's *t* test was performed between the wild-type and the respective mutant strains using a 95% confidence interval, $^{ns}P > 0.05$; *$P \leq 0.05$; **$P \leq 0.01$, ***$P \leq 0.001$. Dots represent individual biological replicates. Source data are provided as a Source Data file.

with the PAS and GAF domains of FphA. Using the full-length version of FphA or the complete photosensory module for AlphaFold 3 did not result in a change in the interaction site for CryA.

To investigate the interaction between the two light receptors in vivo, we fused each of the two proteins to either half of the yellow fluorescent protein (YFP) to perform a bimolecular fluorescence complementation (BiFC) assay (Fig. 3b). Fluorescence microscopy revealed that CryA and FphA interact in the nucleus. The blue-light receptor LreA was used as a negative control and resulted in no detectable fluorescence signals (Supplementary Fig. 3b). To confirm CryA-FphA interaction, we used a biolayer interferometry (BLI) assay to analyze the binding kinetics of the interaction (Fig. 3c). This in vitro assay is an optical method that measures the interference pattern of white light, which is reflected from a biomolecular layer of the two interacting proteins. One interaction partner is immobilized as a bait on a biosensor via antibody/antigen interaction to form the first biolayer. This biosensor is then immersed in a solution of the binding partner, which acts as the analyte, and alters the optical thickness. The resulting interference pattern depends on the concentration of the analyte, and these changes can be analyzed as binding kinetics, providing accurate measurements of the association and dissociation kinetics between the two interacting partners[65–67]. FphA was heterologously expressed in *E. coli*, and the success of its purification was verified by size-exclusion chromatography (SEC) and SDS–PAGE. The functionality of FphA was determined by spectroscopy (Supplementary Fig. 3c–f). Only the functional dimer of FphA was used for the assay. CryA was used as a bait protein, and several solutions of FphA with increasing concentrations were prepared as the analyte. FphA was illuminated with white light before each experiment. The BLI assay revealed a binding affinity of 0.0682 μM, indicating a strong interaction between the two light receptors, thereby confirming that CryA and FphA physically interact. To determine whether light-induced activation of FphA is necessary for the interaction with CryA, we mutated Cys195, the residue crucial for chromophore binding[47], and repeated the BLI assay with CryA as bait and FphA^C195A at the indicated concentrations as analyte (Supplementary Fig. 4a). The interaction between the two proteins was significantly reduced, resulting in a binding affinity of 0.345 mM, which suggests the importance of the phytochrome chromophore in this protein interaction.

Previous RNAseq experiments after light exposure of *A. nidulans* revealed 1134 differentially expressed genes (DEGs). Of these, only 11 genes were exclusively controlled by blue light, demonstrating that phytochrome is the primary component responsible for light-dependent gene activation[15]. FphA primarily activates the light response by activating AtfA through the HOG pathway, but it can also enter the nucleus to regulate gene expression via histone 3

modification[1,50]. Because CryA and FphA primarily interact in the nucleus, we hypothesized that CryA negatively regulates phytochrome function by inhibiting H3K9 acetylation. To test this hypothesis, we performed chromatin immunoprecipitation (ChIP) and compared the abundance of modified histones in the wild-type with the Δ*cryA* strain. We used antibodies against histone (H3) and antibodies against the acetylated H3 (H3K9) for the ChIP with *actA* as a control and *ccgA* as the target gene of FphA (Fig. 3d). In the deletion strain, the abundance of acetylated *ccgA* chromatin was significantly enriched in comparison to the input control, demonstrating that CryA negatively affects the acetylation and thereby *ccgA* activation by phytochrome.

To test if CryA is only involved in the red-light response, we studied the expression of several genes in a *cryA*- and an *fphA*-deletion strain (Fig. 3e). We selected genes that are weakly responsive to blue light and strongly expressed after red-light illumination (*ccgA* and *ccgB*). In addition, we selected *AN8930*, a red light-responsive gene that did not show any activation after blue-light illumination according to RNAseq data, to exclude any involvement of the blue-light receptor LreA[15]. All genes were significantly upregulated in WT after red-light exposure. This increase was completely abolished in the *fphA*-deletion mutant, indicating phytochrome as the key positive factor for expression. The *cryA*-deletion mutant showed a significantly stronger induction of expression after red-light illumination, demonstrating that CryA is a negative regulator of the phytochrome response. Surprisingly, *cryA* deletion also resulted in a significant increase in the expression of *ccgA* and *ccgB* after blue-light illumination, suggesting either an indirect involvement of the White-Collar complex or that CryA also has inhibitory effects on the blue-light response of FphA. FphA can also act as blue-light receptor[1,15]. Because CryA did not interact with LreA the latter possibility appears more likely. As a control, we generated a Δ*cryA*/Δ*fphA* double-deletion strain to analyze if the expression of the light-dependent genes is primarily controlled by FphA and not by CryA (Supplementary Fig. 4b). The double mutant displayed no noticeable upregulation of gene expression upon light treatment and a significant downregulation of the developmental transcription factor *brlA*, indicating that the transcriptional activation is primarily due to phytochrome.

## CryA negatively regulates the oxidative stress response

*A. nidulans* uses several conserved phosphorelay systems and MAPK pathways for signal transduction in response to external stress stimuli[68,69]. The HOG pathway with AtfA as the terminal transcription factor plays a key role in the adaptation to environmental stresses[53,70–72]. Recently, the activation of expression of *cryA* was shown to depend on AtfA when exposed to menadione (MD)[53]. Therefore, we tested the effect of different reactive oxygen species on the expression

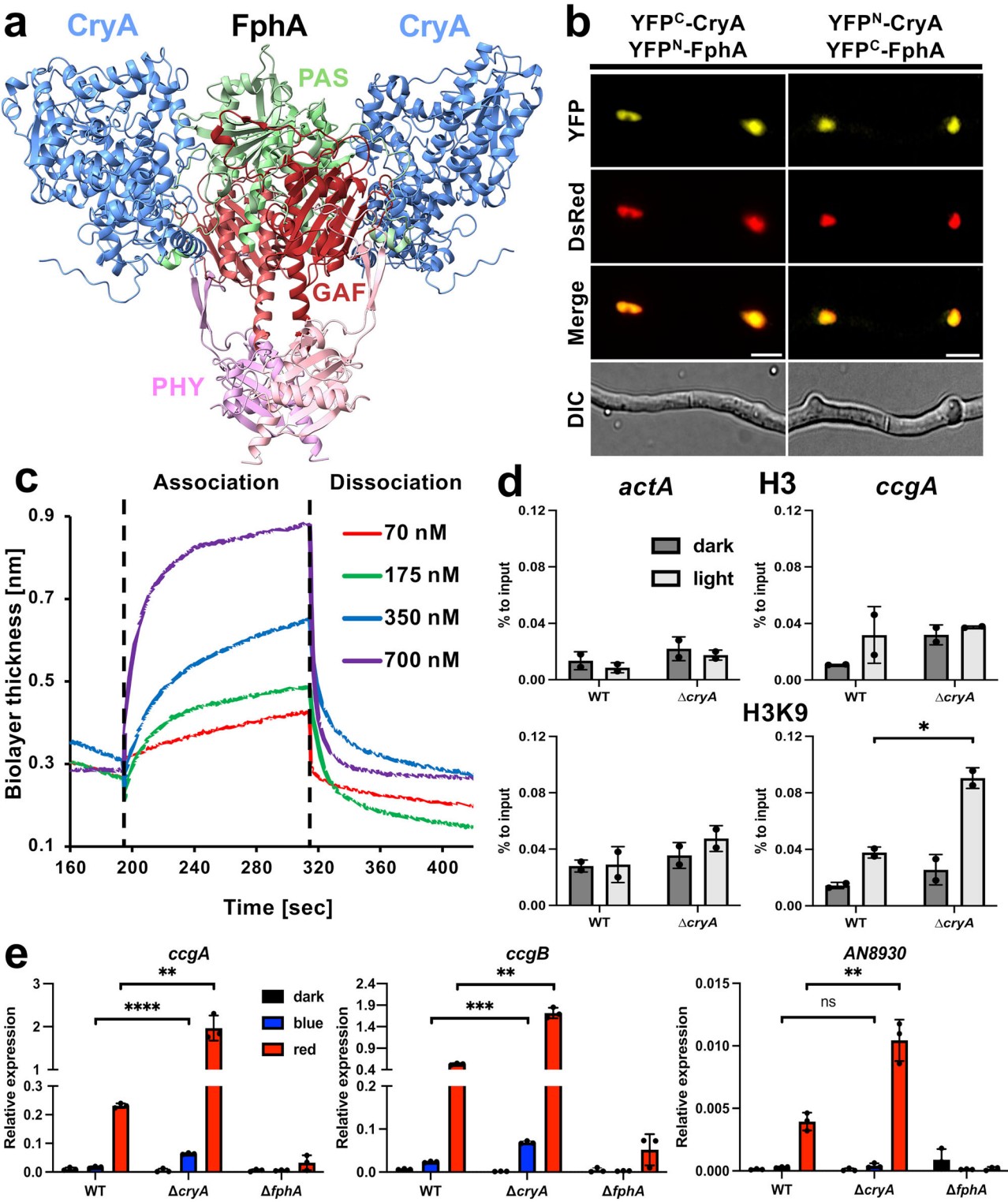

level of *cryA* (Fig. 4a). Exposure to 10 mM $H_2O_2$ or 0.05 mM MD resulted in a significant upregulation of the *cryA*-expression levels. The relative expression of *cryA* after ROS treatment was comparable to that after light induction. Next, we analyzed the phenotypic changes for the *cryA* mutants with different stress stimuli. For this experiment, we compared wild-type, the *cryA*-deletion strain, the *cryA*-overexpression strain, and an *fphA*-deletion strain. The strains were grown on minimal media supplemented with 2% threonine for 5 days at 30 °C in white light without any supplements (control) or with either 1 M NaCl, 10 mM $H_2O_2$, or 0.05 mM menadione (Fig. 4b). Both NaCl and MD inhibited

the growth of all strains, although the Δ*cryA* strain exhibited increased growth and conidiation on MD. In addition, only the Δ*cryA* strain displayed visible growth on 10 mM $H_2O_2$, suggesting that CryA controls the resistance against oxidative stress agents.

One of the multilayered oxidative stress responses of *A. nidulans* to external ROS utilizes the HOG pathway with AtfA as transcription factor to control the expression of several catalases, glutathione- and thioredoxin reductases[71,73]. Modifications of the activity of AtfA on the protein level might balance this stress response. Therefore, we fused both proteins N-terminally to either half of YFP to perform BiFC assays

**Fig. 3 | CryA negatively regulates the red-light receptor phytochrome.**
**a** AlphaFold 3 was used to predict the interaction between the phytochrome dimer and CryA (light blue). For better visibility, only the conserved PAS (green), GAF (red), and PHY (pink) domains of the photosensory domain of phytochrome without the N-terminal extension are visualized. **b** BiFC assay with CryA and FphA. Strains were incubated overnight at 25 °C in minimal media with 2% threonine to stimulate promoter activity. DsRed C-terminally fused with the NLS of StuA was used for nuclei visualization[113]. The scale bar represents 5 μm. The experiment was repeated at least 10 times with similar results. **c** Biolayer interferometry analysis with purified CryA with 6xHis-tag and FphA with Strep-tag. CryA was immobilized as bait with a concentration of 30 μg/ml on a Nickel-nitrilotriacetic acid (NTA) biosensor. Phytochrome was prepared at the indicated concentrations, and association/dissociation kinetics were recorded to calculate the $K_d$ value with the BLI software of the system. All protein solutions were made up in BLI buffer. For the analysis, a global fit with a step correction for the start of the association and dissociation was selected. As a reference, the BLI buffer without protein was used. The experiment was repeated at least 3 times with similar results. **d** For the ChIP analysis, four biological replicates of the respective strains for each condition were incubated overnight in 50 ml minimal media at 37 °C. Before fixation with

formaldehyde, the four replicates were illuminated (light) or kept in darkness (dark) and combined into two replicates. ChIP was performed with antibodies against histone 3 (top) and acetylated lysine 9 of histone 3 (bottom). qRT-PCR was used for *ccgA* and the control gene *actA*. The abundance of immunoprecipitated product was compared against the total chromatin input. Error bars represent the standard deviation of two biological and two technical replicates. For statistical analysis, an unpaired, parametric two-tailed Student's *t* test was performed, using a 95% confidence interval, *$P \leq 0.05$. Dots represent individual biological replicates. **e** Expression analysis of different red-light-regulated genes in wild-type and Δ*cryA* and Δ*fphA* strains. $1.5 \times 10^8$ spores of the respective strains were incubated for 16 h in 10 ml minimal media with 2% glucose at 37 °C completely in the dark or were illuminated for 15 min with blue- or red light (200 μmol photons/m²*s²) before harvesting the mycelia under green safety light. For the qRT-PCR *h2b* was used as a housekeeping gene for normalizing the expression level. Error bars represent the standard deviation of three biological replicates and two technical replicates. For statistical analysis, an unpaired, parametric two-tailed Student's *t* test was performed, using a 95% confidence interval, $^{ns}P > 0.05$; **$P \leq 0.01$; ***$P \leq 0.001$; ****$P \leq 0.0001$. Dots represent individual biological replicates. Source data are provided as a Source Data file.

to analyze a potential interaction between CryA and AtfA (Fig. 4c). Indeed, the proteins interacted in the nucleus. To further investigate the effect of CryA on the AtfA-dependent stress response, we performed expression analyses of several genes under the control of AtfA that are regulated by oxidative stress (Fig. 4d). All genes analyzed were activated after exposure to $H_2O_2$ in wild-type. The Δ*cryA* mutant showed a significantly stronger upregulation of catalase *catB*, and a significantly lower induction of the expression of all other genes compared to wild-type. The survival of the *cryA*-deletion strain on high $H_2O_2$ stress can be attributed to the stronger expression of *catB*, which is primarily active in vegetative hyphae and a primary antioxidant against $H_2O_2$[71,74,75]. These results suggest that CryA controls ROS-related gene expression by interacting with AtfA, thereby fine-tuning the oxidative stress response.

## Cys[42] in the N-terminal extension of CryA is required for ROS-dependent re-localization from nuclei to mitochondria

When we tested the response of *A. nidulans* to hydrogen peroxide, we found that the localization of CryA varied. Under normal conditions, CryA resided in nuclei, but exposure to 10 mM $H_2O_2$ led to an immediate export of CryA from the nucleus and shuttle to the mitochondria within seconds (Fig. 5a and Supplementary Movie 1). This was surprising because protein sequence analysis did not predict a canonical mitochondrial import motif. Typically, cryptochromes have nuclear localization signals (NLS) in the C-terminal region, whereas CRY-DASH members have additional N-terminal motifs for mitochondrial or plastidial import[54,76,77]. We therefore analyzed the NTE of CryA, which is highly disordered and is only present in other eukaryotic CPD photolyases and CRY-DASH members (Fig. 1a). Sequence alignment of CryA with Cry2 from *A. thaliana* revealed that the 70 amino acids of the NTE are unique to CryA, and the consensus region of both proteins starts after Met71 of CryA (Fig. 5b). To investigate the function of the NTE, we generated mutants of CryA lacking either the first 20, 40, or 60 amino acids and fused the protein N-terminally to GFP. CryA is localized exclusively to mitochondria for all truncation variants (Fig. 5c). These results suggest that the first 20 amino acids of the NTE of CryA are important for nuclear import and retention. MD treatment did not induce the shuttle.

As a control, we tested whether the localization of AtfA was changed in the *cryA*-deletion strain (Supplementary Fig. 5). In wild-type, AtfA is only found in the nucleus[78], and we did not observe any changes in localization in the Δ*cryA* strain. Additionally, we analyzed the interaction between CryA and AtfA following treatment with light or $H_2O_2$ but did not observe any nuclear export or shuttling to mitochondria. This system cannot be used to study the impact of $H_2O_2$ on the interaction because YFP is very sensitive to the reagent and loses its fluorescence[79].

We hypothesized that the $H_2O_2$-induced CryA shuttle to mitochondria results from a conformational change of CryA. This raised the question how CryA is able to sense exogenous ROS. Cysteine and methionine can be oxidized by ROS, and hence are candidates to induce structural changes of proteins[80–82]. The NTE of CryA indeed contains a cysteine at position 42 with the potential to form a disulfide bond with cysteine 387 or cysteine 391, which are close enough for such a reaction as predicted with AlphaFold 3 (Fig. 6a). Indeed, a mutated variant where Cys42 was changed to alanine (CryA[C42A]) remained in nuclei after treatment with hydrogen peroxide, indicating that Cys42 is necessary for the movement of CryA to mitochondria (Fig. 6b).

Since the Δ[NTE]CryA truncation mutant localized exclusively to mitochondria, and the CryA[C42A] mutant to nuclei, we tested how these different localizations affect stress responses. We introduced and overexpressed both mutant versions of CryA in the Δ*cryA* strain and compared their growth on media containing $H_2O_2$ (Fig. 6c). Interestingly, the CryA[C42A] mutant behaved like the deletion mutant on higher concentrations of hydrogen peroxide. In contrast, the overexpression of Δ[NTE]CryA reinstated susceptibility to $H_2O_2$. These results were surprising because we anticipated that the regulatory effect of CryA was exclusively connected to nuclear processes. We expected the Δ[NTE]CryA strain to phenocopy the *cryA*-deletion strain and the cysteine variant to behave identically to wild-type. Since this was not the case, we examined the effect of $H_2O_2$ on the expression levels of several oxidative stress genes (Fig. 6d). Treatment with hydrogen peroxide led to strong upregulation of *catB* and *glrA* in the CryA[C42A] strain. Conversely, the mitochondrial variant of CryA exhibited a weaker response to hydrogen peroxide. To expand on this, we repeated the experiment and compared the growth and expression profiles of both mutant strains on menadione with the wild-type and the Δ*cryA* strain (Supplementary Fig. 6a, b). In addition to catalases, thioredoxin, and glutathione genes, we included *prxA* and *cetJ* because they have been shown to act as antioxidants against menadione[53,83]. Interestingly, all CryA variants were more resistant to menadione and displayed significant upregulation of gene expression compared to the wild-type. We conclude that CryA localization appears to be related to cell signaling mechanisms that activate resistance genes against various reactive oxygen species. This presents a novel mechanism by which *A. nidulans* balances stress responses.

## Discussion

Transcriptional gene activation is a common mechanism for cells to respond to changing external and internal environments. Positive feedback loops guarantee quick reactions, and negative feedback loops counteract the responses, prevent harmful overstimulation, and thereby help to control gene activity in space and time. Examples of

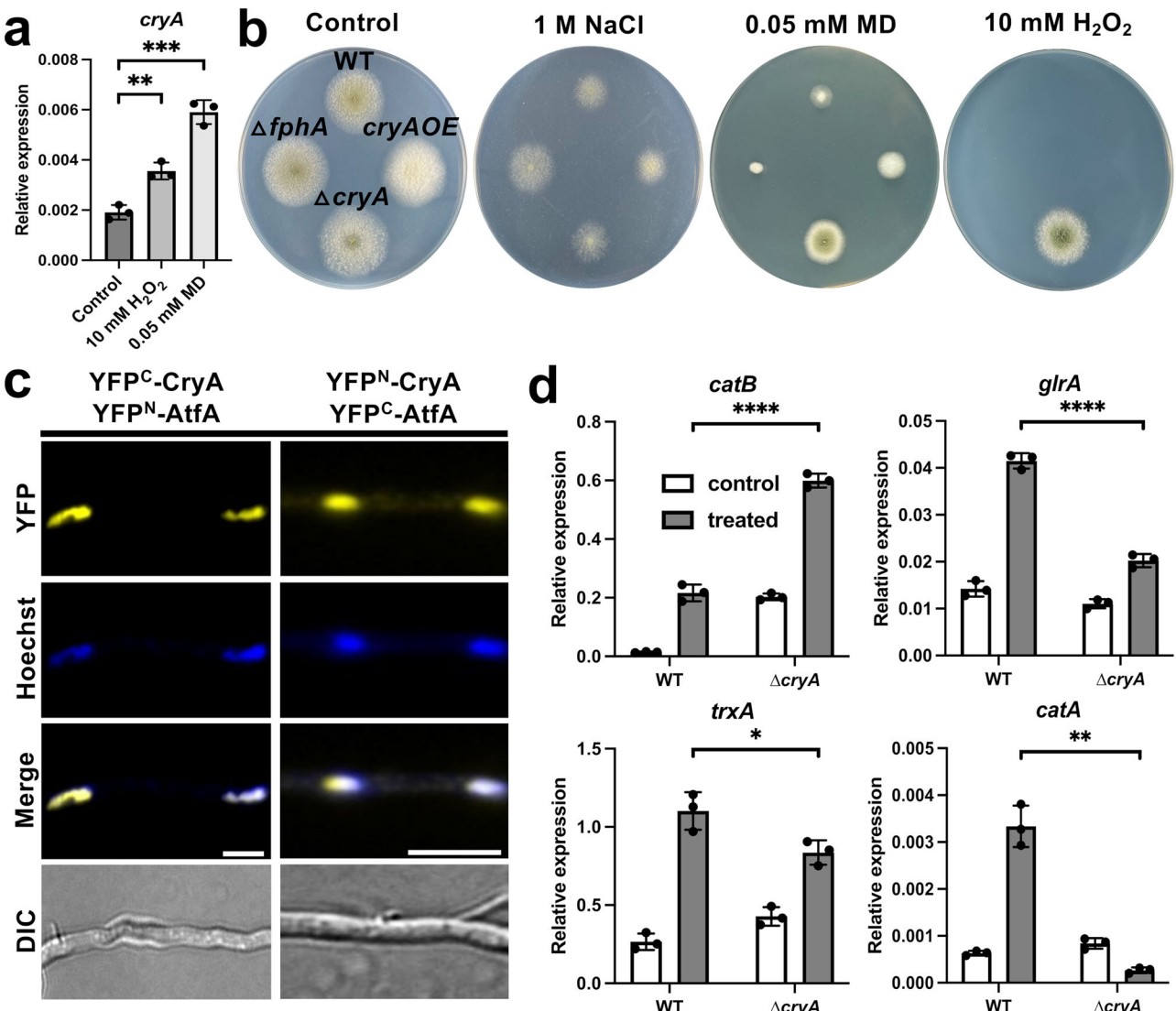

**Fig. 4 | CryA interacts with the stress-activated transcription factor AtfA to regulate the oxidative stress response. a** Analysis of the *cryA*-expression level after exposure to different oxidative stress conditions. $1.5*10^8$ spores of wild-type were incubated for 16 h in 10 ml minimal media at 37 °C in the dark. Before harvesting, the mycelia were transferred to pre-warmed liquid minimal media without supplements (control, light gray), 10 mM $H_2O_2$ (gray), or 0.05 mM menadione (MD, dark gray) under green safety light and incubated for 25 min. For the qRT-PCR, *h2b* was used as a housekeeping gene for normalizing the expression level. Error bars represent the standard deviation of three biological and two technical replicates. For statistical analysis, an unpaired, parametric two-tailed Student's *t* test was performed, using a 95% confidence interval, **$P \le 0.01$, ***$P \le 0.001$. Dots represent individual biological replicates. **b** Comparison of WT, the *cryA*-deletion strain (Δ*cryA*), the *fphA*-deletion strain (Δ*fphA*), and the *cryA*-overexpression strain (*cryA*OE) exposed to either 1 M NaCl, 0.05 mM MD, or 10 mM $H_2O_2$. In total, 1000 spores were used for each colony with 8 replicates per condition. The plates were incubated on minimal media supplemented with 2% threonine instead of glucose for 5 days at 30 °C in white light (200 μmol photons/m²·s²). **c** BiFC assay of N-terminally tagged CryA and AtfA. Strains were incubated overnight at 30 °C in minimal media with 2% glycerol to induce the *alcA* promoter. Nuclei were stained with HOECHST. The scale bar represents 5 μm. The experiment was repeated at least ten times with similar results. **d** Expression analysis of genes responding to oxidative stress in wild-type and the Δ*cryA* strain. $1.5*10^8$ spores of the respective strains were incubated for 16 h in 10 ml minimal media at 37 °C, completely in the dark. Before harvesting, the mycelia were transferred to pre-warmed liquid minimal media without supplements (control, white) or 10 mM $H_2O_2$ (treated, gray) under green safety light and incubated for 25 min. For the qRT-PCR *h2b* was used as a housekeeping gene for normalizing the expression level. Error bars represent the standard deviation of three biological and two technical replicates. For statistical analysis, an unpaired, parametric two-tailed Student's *t* test was performed, using a 95% confidence interval, *$P \le 0.05$, **$P \le 0.01$, ****$P \le 0.0001$. Dots represent the individual biological replicates. Source data are provided as a Source Data file.

environmental responses are light- and/or stress-induced genes in fungi, where receptors and signal transduction cascades have been described in great detail. However, mechanisms for fine-tuning and signal-cascade control are less well studied. Here, we analyzed such interplay in *A. nidulans*. This organism senses red light with fungal phytochrome and uses the HOG stress pathway for red-light signal transduction with AtfA as the final transcriptional regulator[1,73]. We discovered that the two pathways not only share parts of the signaling cascade for gene activation, but that the cryptochrome CryA acts as a negative regulator for both pathways. The *cryA* gene is activated by light and by oxidative stress, and the negative feedback relies on physical interaction of CryA with the photoreceptor FphA and the transcription factor AtfA, thereby balancing both responses. We are going to discuss (i) the role and mechanism of CryA in light regulation, (ii) its dark functions in development regulation, and (iii) the role and mechanism of CryA in stress adaptation.

Light-dependent activation of genes should be transient and therefore needs to be negatively regulated to enable precise control at

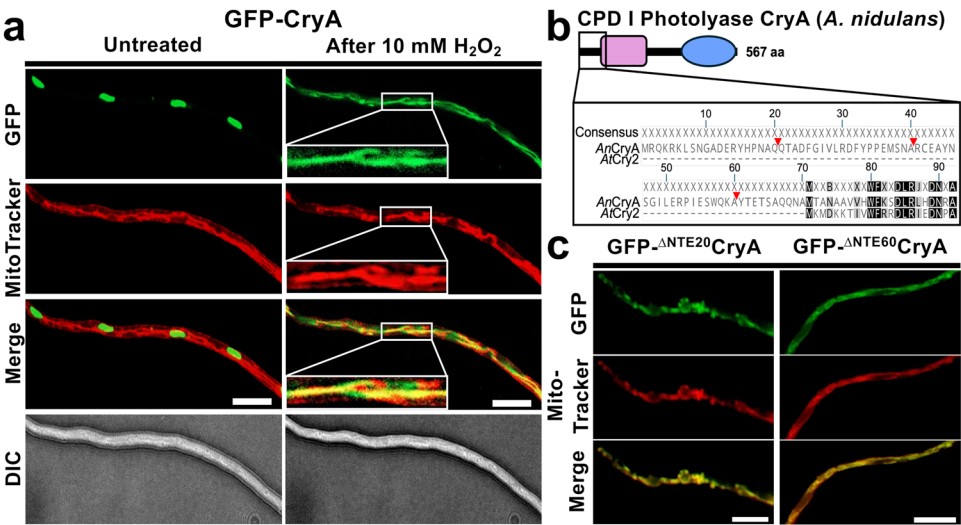

**Fig. 5 | ROS-dependent localization of CryA depends on the N-terminal extension. a** Fluorescence microscopy to localize CryA before and after exposure to 10 mM $H_2O_2$. Mycelia was incubated overnight at 25 °C in minimal media with 2% threonine to induce the promoter activity. MitoTracker staining was used to visualize mitochondria. The scale bar represents 10 μm. The experiment was repeated at least ten times with similar results. **b** Comparison between the N-termini of CryA and Cry2 from *A. thaliana*. Geneious Prime 2024 was used for pairwise alignment with a global alignment with free end gaps, a Pam-matrix of 100, and gap costs of 10/1. Red triangles mark the chosen lengths for the truncation mutants. **c** Fluorescence microscopy of a strain expressing truncation mutants of the N-terminal extension of CryA. Strains were incubated overnight at 25 °C in minimal media with 2% threonine to induce promoter activity. MitoTracker staining was used to visualize mitochondria. The scale bar represents 10 μm. The experiment was repeated at least ten times with similar results.

varying light levels. Hence, the negative action of CryA could be considered as a photoadaptation process, given that *cryA* is light-induced and acts in a negative-feedback loop on the expression of other light-induced genes (Fig. 7). Unexpectedly, unlike in *N. crassa*, where photoadaptation is modulated by Vivid (VVD) and Frequency (FRQ) as negative elements, CryA did not affect the late light response. Normally, FRQ recruits casein kinase to phosphorylate and inactivate the WCC, and VVD disrupts the homodimerization of the WCC by competitive interaction with WC-1 at the later stages of photoadaptation[24,51]. In *A. nidulans*, CryA appears to behave similarly to ENV1 from *Trichoderma reesei*, a VVD-like PAS/LOV domain containing inhibitor primarily of the early light response[84,85]. Furthermore, the CryA homologs Cry1 and Phr1 are connected to the Blr1-controlled photoresponse in *T. atroviride*, although the precise mechanism is still unclear[39,40].

CryA interacts with FphA and with AtfA to regulate the photoresponse which raises the question why one pathway needs to be controlled at two places. Phytochrome resides in the cytoplasm and activates the HOG pathway by interacting with the phosphotransfer protein YpdA, relaying the signal through the MAP kinase cascade to activate AtfA[48]. Besides this cytoplasmic function, a fraction of FphA has also a nuclear function in chromatin remodeling[49,50]. Indeed, CryA influenced histone acetylation. Through the interaction of CryA with AtfA, the transcriptional-activation activity of FphA is also controlled because AtfA is downstream of FphA. In addition, the interaction of CryA with AtfA not only affects the light response but also the oxidative-stress response. Hence, CryA regulates the two modes of action of FphA and in addition allows integration of light-independent stress signaling. The control of the activity of FphA and AtfA appears to occur through protein-protein interactions. However, it remains open if the regulatory effect is due to conformational changes of the target protein or steric hindrance of other interactions, or if the interaction is accompanied by changes in kinase or phosphatase activity of FphA. There is experimental evidence for such interrelationships in plants. For instance, *A. thaliana* CRY1 and CRY2 interact with the transcription factors PIF4 and PIF5 to compete with phytochrome binding[86]. It was also shown that CRY1 and CRY2 interact with phytochrome PhyA. But in this case, cryptochromes were phosphorylated by PhyA[87–89]. Light-responsive physical interactions between PhyA and CRY1, PhyB and CRY2, or PhyB and CRY1 have also been reported, and it was suggested that CRY1 acts as a negative regulator of the red-light response[87,88,90].

Besides the control of light-regulated genes, some of which with roles in development, CryA controls development in a light-independent way. We observed significant downregulation of several asexual transcription factors in the *cryA*-overexpression strain, which resulted in white (fluffy) colonies without conidia. These results are in line with findings in *A. thaliana* with CRY2, which is also active in darkness and can interact with PhyB despite lacking the CCE domain for protein interaction[90,91]. In addition, although the mechanism behind the dual function and how these proteins can interact and regulate without C-terminal extension remains to be elucidated, fungal photolyases with cryptochrome-like functions seem to be a common trait in the fungal kingdom[33,39–42]. Although CryA appears bioinformatically and structurally as a photolyase, there are several cases of photolyases with cryptochrome activity outside the fungal kingdom[36–38]. It remains unclear whether the developmental effects in *A. nidulans* can all be explained by the interaction with FphA and/or AtfA, but it appears conceivable that CryA also interacts with other cellular targets in an unknown way. A link between the observed phenotypes could also be the role of CryA in the control of the redox state of the cell. This will be the last phenomenon discussed here.

Exposure of *A. nidulans* to oxidative stress induced the upregulation of *cryA* along with anti-stress genes. CryA then interacts with AtfA to fine-tune the activation of those anti-stress genes. An intriguing finding in our work is the observed rapid shuttle of CryA from nuclei to mitochondria. The N-terminal extension was critical for this translocation, suggesting a novel way for cryptochrome conformational changes accompanied by changes in its subcellular localization. This raises two interesting questions: (i) what is the function of CryA at mitochondria and (ii) how the NTE controls the shuttle. Regarding the first point, CryA might be connected to blue and UV light, causing high intracellular ROS levels that can damage not only nuclear but also mitochondrial DNA. Therefore, CryA could be sent as an "emergency weapon" to mitochondria to catalyze DNA repair and prevent mitochondrial DNA fragmentation[77,92]. However, it could also be that CryA plays a regulatory role in mitochondria. In *Drosophila*, it was shown that the mitochondrial respiratory chain impinges on CRY activity.

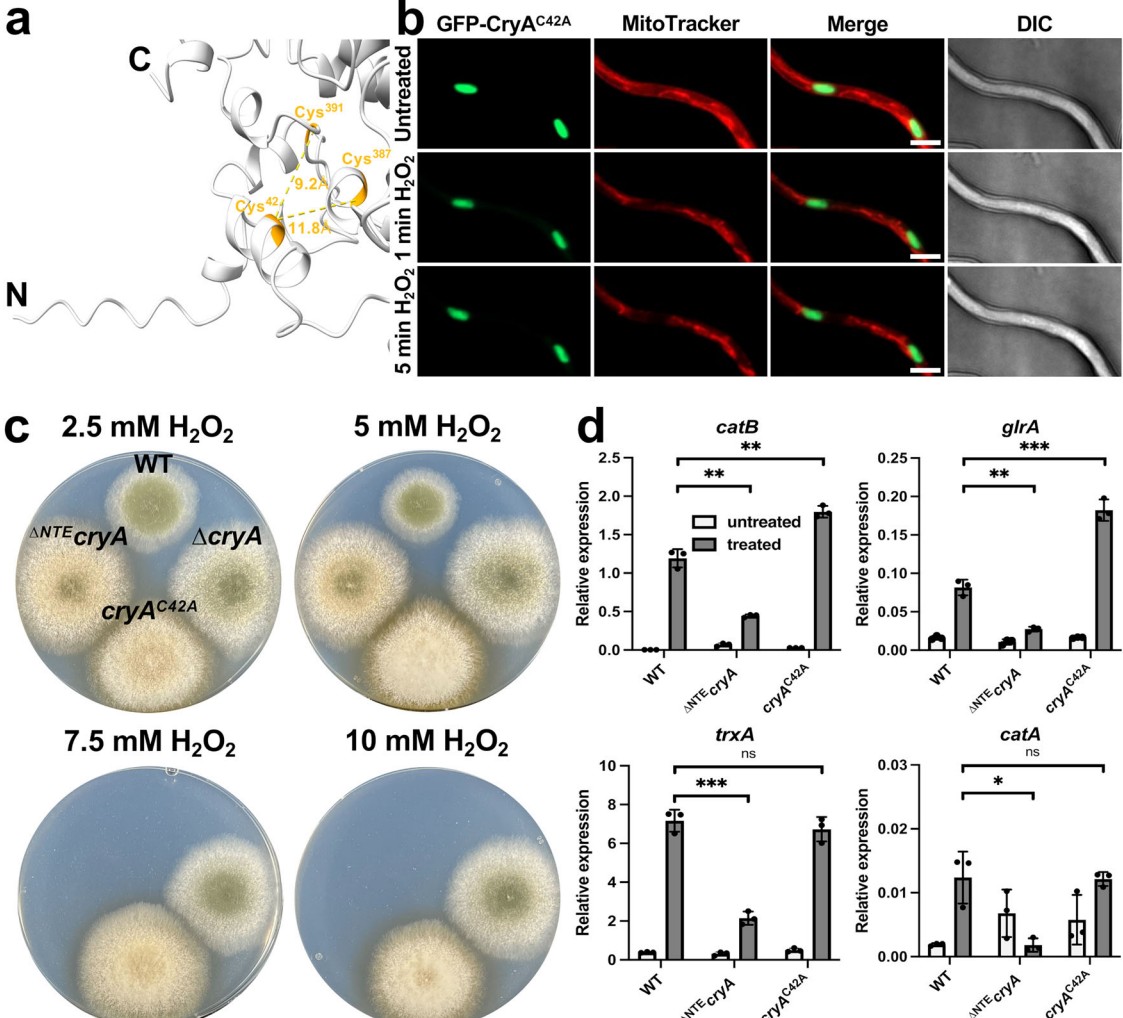

**Fig. 6 | A conserved cysteine in the NTE determines the subcellular localization of CryA. a** AlphaFold 3 and ChimeraX were used to visualize cysteine (orange) in the NTE. The distance between the cysteine is given in Ångström. **b** Fluorescence microscopy to localize CryA[C42A] before and after exposure to 10 mM $H_2O_2$. Spores were incubated overnight at 25 °C in minimal media with 2% threonine to induce promoter activity. MitoTracker staining was used to visualize mitochondria. The scale bar represents 5 µm. The experiment was repeated at least ten times with similar results. **c** Comparison of wild-type colonies with the *cryA*-deletion, the Δ*cryA* strain re-complemented with an overexpression mutant of the N-terminal truncation mutant of CryA ([ΔNTE]*cryA*), and re-complementation of the Δ*cryA* strain with the cysteine mutant (*cryA*[C42A]). 1,000 spores were grown for 5 days at 30 °C on minimal media with 2% threonine in light (white light with 200 µmol photons/m²*s²) with the

indicated concentrations of $H_2O_2$. **d** Expression analysis of genes responding to oxidative stress in wild-type, the [ΔNTE]*cryA* strain, and the *cryA*[C42A] strain. In total, 1.5*10^8 spores of the respective strains were incubated for 16 h in 10 ml minimal media at 37 °C completely in the dark. Before harvesting, the mycelia were transferred to pre-warmed liquid minimal media without supplements (control, white) or 10 mM $H_2O_2$ (treated, gray) under green safety light and incubated for 25 min. For the qRT-PCR, *h2b* was used as a housekeeping gene for normalizing the expression level. Error bars represent the standard deviation of three biological and two technical replicates. For statistical analysis, an unpaired, parametric two-tailed Student's *t* test was performed, using a 95% confidence interval, [ns]$P > 0.05$; *$P \leq 0.05$, **$P \leq 0.01$, ***$P \leq 0.0001$. Dots represent the individual biological replicates. Source data are provided as a Source Data file.

Inhibition of mitochondrial complex III and complex V resulted in increased intracellular ROS production and impeded CRY regulation of the circadian clock, suggesting that CRY is somehow connected to the mitochondrial respiratory chain[93]. It is therefore likely that CryA responds to changes in the mitochondrial redox cycle and promotes survival for mitochondria, as described for other CPD I-II photolyases or cryptochromes[77,101].

The response of CryA to external ROS may be related to the observed developmental phenotypes of the *A. nidulans* Δ*cryA* or the overexpression mutant. We therefore propose that CryA is involved in the oxidative stress response, which is a tightly controlled system of several multilayered pathways and transcription factors that control antioxidants such as catalases. These systems complement each other at different stages of the life cycle of *A. nidulans*, with CatA being primarily active in conidia and CatB in

vegetative hyphae[70,71,73–75,95]. GlrA, a glutathione reductase, and TrxA, the thioredoxin A, are involved in balancing intracellular redox levels[96,97]. All these antioxidant genes were differentially expressed in the *cryA*-deletion mutant, probably through the action of AtfA. In addition, there are other transcription factors that are critical for the oxidative stress response, such as the bZIP protein NapA, which shuttles to the nucleus after oxidation by ROS[75,98,99]. NapA then stimulates catalase B, the thioredoxin system, and glutathione reductase[100]. This provides a parallel signaling pathway independent of the HOG pathway to confer resistance to ROS. Therefore, we propose that the shuttle of CryA in high ROS environments is a molecular mechanism to (i) protect the mitochondria from ROS-induced DNA damage and (ii) remove the negative effect of CryA in nuclei to induce stress responses via AtfA and possibly NapA (Fig. 7).

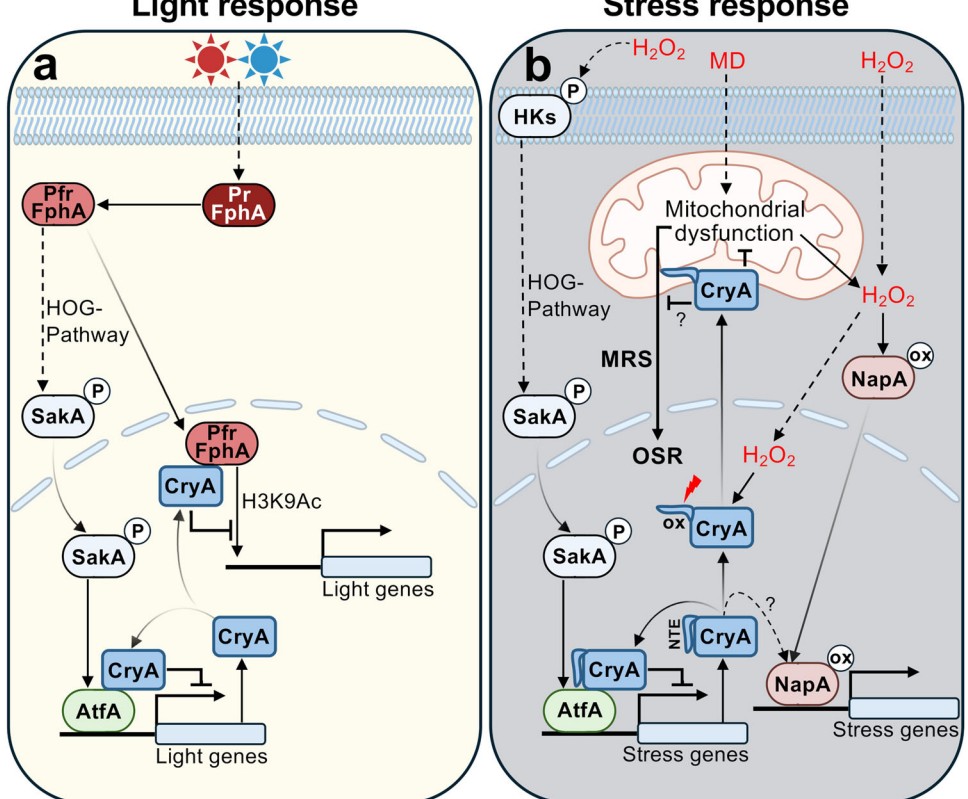

**Fig. 7 | Model of CryA-regulated light and stress responses in *A. nidulans*.**
**a** Exposure to red or blue light activates phytochrome conformational changes from the inactive (Pr) to the active form (Pfr). Photo-activated FphA triggers the HOG pathway, which transduces a signal via SakA (HogA) into the nucleus to the transcription factor AtfA, which induces the transcription of light-responsive genes. In addition, FphA can shuttle into the nucleus to activate chromatin remodeling of light-responsive genes. CryA is expressed, accumulates in the nucleus, and inhibits the function of FphA and AtfA by physical interaction, thereby negatively regulating the light response in a negative-feedback loop. **b** Exposure to ROS such as $H_2O_2$

triggers the HOG pathway with AtfA and induces the shuttle of NapA into nuclei to activate the oxidative stress response. CryA is expressed and modifies the oxidative stress response, fine-tuning the activation of AtfA and possibly NapA. Exposure to $H_2O_2$ induces the shuttle of CryA from nuclei to mitochondria by oxidizing cysteine 42 in the NTE, resulting in a conformational change. CryA potentially disrupts menadione (MD)-induced mitochondrial dysfunction and $H_2O_2$ production, thereby preventing mitochondrial retrograde signaling (MRS) and activation of the oxidative stress response (OSR).

The next interesting point was the role of the NTE in determining the subcellular location. Whereas the full-length protein resided in nuclei and shuttled to mitochondria in the presence of $H_2O_2$, the protein lacking the NTE was found exclusively at mitochondria. One possibility to explain this phenomenon was that the NTE is proteolytically processed. However, in our experiments, CryA was fused N-terminally to GFP, and cleavage of the NTE would hence release GFP to the cytoplasm or nucleus, which we did not observe. A second possibility was that nuclear CryA is degraded, and newly synthesized CryA is targeted to mitochondria. This mechanism also seems unlikely because of the almost instantaneous shuttling of the protein after exposure to $H_2O_2$. Therefore, it appears likely that the NTE somehow blocks a mitochondrial targeting sequence or prevents interaction with nuclear export proteins. Although there are some other cases of CPF proteins with the ability to localize to mitochondria, the underlying mechanism remains elusive[77,92–94,101]. The rice CPD II photolyase repairs UV-B light-induced mitochondrial and plastidal DNA damage and can localize to mitochondria, plastids, and the nucleus[77]. Interestingly, the KRK motif relevant for nuclear localization is also present in the NTE of CryA, and the removal of the first 20 amino acids prevented nuclear import. In addition, the amino acid sequence MHWFR, which is relevant for the photoreactivation of mitochondrial DNA in *Saccharomyces cerevisiae*, is predicted to be an MTS for CPD I photolyases[94]. A similar motif is present in CryA immediately after the NTE at position 78-82 (VHWFK). Taken together, it appears that the

contact with ROS triggers a conformational change of the NTE, resulting in exposure of the MTS followed by nuclear export and translocation to mitochondria. This conformational change could be induced by redox-sensitive amino acids such as cysteine and methionine. Oxidation of these amino acids, specifically the formation of disulfide bridges in cysteine and sulfoxide in methionine, has been shown to induce conformational and functional changes in proteins, effectively functioning as a regulatory switch for cellular responses.[80–82] Mutagenesis of Cys42 in CryA indeed abolished the shuttle after $H_2O_2$ treatment, supporting the hypothesis that this amino acid is crucial for nuclear retention and mitochondrial transport.

Investigating the function of a protein in two different locations within a cell is usually difficult, since deletion of the gene affects both functions. In the case of *A. nidulans* cryptochrome, the constitutive localization of the truncated CryA version to mitochondria offers the opportunity to analyze ROS functions outside the nucleus, while the cysteine mutation allows for the same analysis in the nucleus. We hypothesized that the "nucleus-only" variant would behave like wild-type and that the truncation mutant would mirror the ΔcryA phenotype. However, the opposite was true. Compared to the full deletion of *cryA*, the "mitochondrial-only" CryA strain exhibited susceptibility to $H_2O_2$, and expression analyses revealed a significantly lower induction of all antioxidant genes. However, the strain still exhibited resistance to menadione, as supported by expression analyses: most antioxidant

genes displayed significant upregulation after menadione treatment. Unexpectedly, the ΔcryA strain displayed a stronger gene activation, suggesting that the truncated CryA variant inhibits the nuclear stress response despite residing at the mitochondria, suggesting mitochondrial-nuclear communication. In *Schizosaccharomyces pombe*, the NapA homolog Pap1 is activated by a mitochondrial-nuclear retrograde pathway, which increases efflux activity to balance redox levels[102]. Therefore, CryA may modulate the activity of AtfA and NapA in a subcellular localization-dependent manner to regulate intracellular redox levels, preventing mitochondrial dysfunction, and fine-tune the stress response (Fig. 7). However, the truncated CryA variant localizes exclusively to the mitochondria. In contrast, $H_2O_2$-induced re-localization of the full-length CryA protein appears to only partially relocate to mitochondria. In contrast, the cysteine-mutated strain tolerated high $H_2O_2$ stress and exhibited a similar expression profile to the full deletion strain. Consistently, the strain remained resistant to menadione; however, gene expression analysis revealed only weak upregulation compared to the wild-type. In addition, CryA's molecular mechanism with menadione appears to differ from its mechanism with hydrogen peroxide. The cysteines in the ROS sensor Yap1 in *S. cerevisiae* are crucial for activation upon contact with $H_2O_2$ and do not respond to other ROS[103,104]. Therefore, we propose that Cys42 is not only relevant for the localization but also for the functionality of CryA upon perception of $H_2O_2$. This explains the phenocopying of the *cryA* deletion.

In conclusion, we have characterized the mechanisms by which CryA negatively regulates light and stress responses in *A. nidulans*. Its role probably includes DNA repair functions, signal cascade control, and gene regulation. Some of these functions appear to depend on structural changes in the protein resulting from cysteine oxidation, followed by re-localization of the protein. Therefore, *A. nidulans* CryA is involved not only in light-dependent regulation but also acts as a sensor for reactive oxygen species.

## Methods

### Strains and culture conditions
Supplemented Minimal Media (MM) or complete media (CM) was used to cultivate *A. nidulans* strains at 37 °C[105]. *E. coli* strain BL21 DE3 was used for protein expression, and Top10 cells were utilized for plasmid replication. Strains are listed in Supplementary Table 2.

### Creation of transgenic microorganisms
Generation and transformation of *A. nidulans* protoplasts was performed by PEG/CaCl₂ transformation[106]. In total, 5 µg of total plasmid DNA was used for transformation. Transformants were incubated at 37 °C for 3 days on MM media with the addition of 1 M sorbitol. Generation and transformation of *E. coli* was performed by heat shock and selection through antiobiotics[107].

### Plasmid construction
All plasmids are listed in Supplementary Table 3 and oligonucleotides for PCR in Supplementary Table 4. All plasmids were generated by using the NEBuilder HiFi DNA Assembly Cloning Kit. For the deletion of *cryA* either 2 kb up- and downstream of the open reading frame of *cryA* were amplified with PCR and attached to a *pyrG* (AO090011000868) cassette as an auxotrophy marker and assembled with a pJET1.2 backbone. The overexpression of *cryA* for phenotypical and molecular analysis was achieved by using the *alcA* promoter with either 2% glycerol for derepression or 2% threonine for overexpression[63,108].

### Heterologous expression in *E. coli*, purification of CryA and FphA, and SDS−PAGE
For protein purification, the pET28a vector (Novagen) for CryA and the pASK-iba3plus vector (IBA Lifescience) for FphA were used. pET28a contains an Isopropyl ß-ᴅ-1-thiogalactopyranoside (IPTG) inducible T7 promoter and either a N- or C-terminal 6xHisTag for purification of the target protein. The pASK-iba3plus vectors contain an anhydrotetracycline (AHT) inducible *tet*-promoter and a C-terminal strep-tag for purification of the target protein. For the purification of FphA, the bacterial heme oxygenase BphO from *Pseudomonas aeruginosa* was assembled in a pACYC vector and co-transformed with FphA. To accomplish the expression of functional FphA, BphO is essential for the degradation of heme to biliverdin, the chromophore of FphA.

For FphA, BL21 DE3 cultures were grown in 2 L liquid LB, supplemented with 100 mM sorbitol, 2.5 mM betaine, and the respective antibiotics, and incubated shaking at 37 °C, 180 rpm until OD₆₀₀ of 0.6 was achieved. To activate expression of BphO, 0.5 mM IPTG was added, and the culture was grown for 120 min at 37 °C. After cooling the culture down to 15 °C, 0.2 µg/ml AHT was added to induce FphA expression. After 16 h at 15 °C and 180 rpm, the culture was centrifuged twice at 9000 rpm at 4 °C for 15 min. The supernatant was removed, and the pellet resuspended in 20 ml buffer A (50 mM Tris-HCl, 300 mM NaCl, 0.05% Tween20, 10% glycerol, 1 mM PMSF, 5 mM DTT, pH 7.8). The cell walls were disrupted by using a high-pressure homogenizer (Emulsiflex-C3), and the pellet was separated by centrifugation for 45 min at 18,000 rpm and 4 °C. In total, 40 µg/ml Avidin was used for 15 min and 4 °C to bind free biotin. The ÄktaPure (Cytiva) was used for protein purification with a 5 ml StrepTrap XT column following the instructions provided by the manufacturer. For elution, buffer A supplemented with 100 mM Biotin was used. For BLI analysis, a size-exclusion chromatography (SEC) with the BLItz-buffer as running buffer was used to remove any aggregated FphA complexes and isolate the functional phytochrome dimer. All steps were performed under green safelight or in darkness.

For CryA, a similar expression protocol was used with the following changes: Induction of CryA was achieved by the addition of 0.5 mM IPTG after cooling down to 15 °C. Buffer A was supplemented with 90 mM Imidazole for wash steps and 500 mM for elution with a pH of 7.4. For purification, a 5 ml HisTrap HP was used following the manufacturer's instructions. Buffer of eluted CryA was exchanged using a PD-10 desalting column (Cytiva) according to the manufacturer's protocol. A Vivaspin ultrafiltration unit (Sartorius) with a 30,000 molecular weight cut off (MWCO) was used to concentrate protein samples. All steps were performed under red safety light or in the dark.

Sodium dodecyl sulfate-polyacrylamide gel electrophoresis (SDS−PAGE) was performed as described[109]. PageRuler 180 kDa pre-stained protein ladder (Thermo Fisher Scientific) was used as a protein marker. The gel was stained with Coomassie Brilliant blue staining (45% Methanol, 10% acetic acid, 44.9% dH₂O, 0.1% Coomassie Brilliant Blue R-250). Destaining was achieved by using Coomassie Brilliant blue destaining solution (40% methanol, 10% acetic acid, 50% ddH₂O).

### Photoreceptor spectroscopy
Absorption spectra were visualized using a JASCO V-750 spectrophotometer (JASCO GmbH), and emission spectra were visualized with the Jasco FP-8300 spectrofluorometer (JASCO GmbH). All measurements were carried out in a 700 µl quartz cuvette at room temperature in red or green safety light.

### Reverse-phase chromatography of the chromophores and spectroscopy
The chromophores from CryA were isolated and determined by boiling the protein at 100 °C for 5 min followed by centrifugation at 15,000×*g* for 15 min at 4 °C[60]. The pellet was removed, and 100 µl of the supernatant was used for subsequent analysis. FAD and FMN standards were prepared identically in water, MTHF was dissolved in 100% Methanol. For reverse-phase high-performance liquid chromatography, a Vanquish Core HPLC System (Thermo Fisher Scientific) with

a Hypersil Gold 150 mm × 4.6 mm, 3 μm C18 column was used. Buffer A1 was composed of 10 mM $Na_2HPO_4$, 45 mM Citric acid, pH 2.4 in $dH_2O$, buffer B1 contained 100% methanol. Equilibration and wash step was performed with 0.5 CV and 10% methanol concentration, elution was performed with 2.5 CV with a continuously increasing methanol gradient to 100%[33].

## Microscopy
Spores were inoculated on microscopy slides with minimal media supplemented with 2% glycerol for interaction analysis or 2% threonine for localization overnight at the indicated temperature. Fluorescence images were captured at room temperature using the AxioImager Z1 and either the Plan-Apochromat 10x/0.5 or 63x/1.4 Oil DIC objective (Zeiss). HOECHST 33342 (Thermo Fisher Scientific) was used for nuclei staining, and MitoTracker™ Red CMXRos (Thermo Fisher Scientific) was used for mitochondria staining. Alternatively to HOECHST staining, DsRed C-terminally fused to the NLS of StuA was used for nuclei visualization[27].

## RNA extraction and reverse transcription quantitative PCR (RT-qPCR)
Unless otherwise stated, $1.5 \times 10^8$ spores were inoculated in 10 ml minimal media and incubated overnight for 16 h at 37 °C. Mycelia was then treated with the respective condition before harvesting. Mortar and pestle were used to grind the mycelia, and RNA was extracted using TRIzol reagent (Invitrogen) and following the manufacturer's instructions. Remaining DNA was digested using the Turbo DNA-free Kit (Invitrogen). qRT-PCR was performed using the Luna Universal One-Step RT-qPCR Kit (NEB) with the CFX Connect Real-Time PCR Detection system (Bio-Rad). In total, 100 ng RNA were used for each reaction mix, and the *h2b* gene (AN3469) was used for normalization. For statistical analysis, the experiments were performed with two technical and three biological replicates.

## Growth assays
For the phenotypic analysis of *cryA* mutants, 1000 spores were inoculated on solid minimal media supplemented with 2% threonine to induce promoter activity. For each condition and strain, 8 replicates were used independently.

## Biolayer interferometry (BLI) assay
The BLItz system (Sartorius) was used for biolayer interferometry assays. Nickel-nitrilotriacetic acid (NTA) coated biosensors were used to immobilize 30 μg/ml CryA tagged with a 6xHis made up in BLItz buffer (137 mM NaCl, 2.7 mM KCl, 10 mM $Na_2HPO_4$, 1 mM $KH_2PO_4$, 0,5% BSA, and 0.04% Tween20, pH 7.4) as bait. FphA in various concentrations made up in BLItz buffer was used as the analyte. For the experiment, the following steps were selected: 30 s baseline in BLI buffer, 120 s loading of CryA to the tip, 45 s baseline step, 120 s association step, 120 s dissociation step[65]. For the calculation of the $K_d$-value, step corrections for the start of the association and dissociation, a global analysis, and a 1:1 binding model were selected. A control of 0 μM FphA was used as a reference for normalization.

## Chromatin immunoprecipitation (ChIP)
ChIP was performed via chemically crosslinking the respective strains in a 50 ml culture for 15 min at 37 °C[110]. Crosslinking was stopped with glycin, and the mycelia was frozen in liquid nitrogen and ground. In total, 100 mg of ground mycelia were sonicated with the Q800R3 sonicator (QSonica) with the following settings: For a total treatment time of 60 min the samples were sonicated in cycles on ice for 2 min ("on") and rested for 1 min ("off") at maximum power. For immunoprecipitation, antibodies against histone 3 (Abcam) and acetylated lysine 9 of histone 3 (Merck), and Protein-G-Agarose (SantaCruz Biotechnology) were used. After washing and reverse crosslinking, the immunoprecipitated DNA was purified with a ChIP cleanup and concentrator kit (Zymo Research). For qRT-PCR, 1 μl of 1:100 diluted input control was used together with 1 μl of purified DNA (1% final concentration of purified DNA/input control). The qRT-PCR was performed with the QuantStudio3 cycler (Fisher Scientific) and the Fast Syber Green Master Mix (Fisher Scientific).

## Bioinformatic and statistical analyses
UniProt (https://www.uniprot.org) and FungiDB (https://fungidb.org/fungidb/app) were used to obtain protein- and genomic sequences. InterPro (https://www.ebi.ac.uk/interpro/) and SUPERFAMILY (https://supfam.org) were used for protein domain predictions. Phylogenetic tree analysis and protein sequence alignments were carried out by Geneious Prime 2024 (https://www.geneious.com). Visualization of quantitative PCR results and statistical analysis was done with Graph-Pad Prism 9.0 (https://www.graphpad.com/features). Microscopic images were processed with ImageJ (https://imagej.net/ij/). Unless specifically noted, each experiment was repeated three or more times independently.

## Protein structure predictions and AlphaFold 3
UniProt protein database was used to obtain the CryA (UniProt ID: Q5BGE3) and FphA (UniProt ID: Q5K039) protein sequences. Alpha-Fold 3 (https://doi.org/10.1038/s41586-024-07487-w) was used to model CryA alone (Date of access: Sep 11, 2024) or together with the full-length or the truncated version of FphA (Date of access: June 18, 2024). Coloring and visualization of protein structures and chromophores from AlphaFold 3 were done with ChimeraX (https://www.cgl.ucsf.edu/chimerax/)[111]. AlphaFill was used together with the protein models generated with AlphaFold 3 to predict chromophores of CryA (https://alphafill.eu) on the Sep 11, 2024. Protein alignment error (PAE) 2D heatmap was generated using PAE viewer (https://pae-viewer.uni-goettingen.de)[112].

## Reporting summary
Further information on research design is available in the Nature Portfolio Reporting Summary linked to this article.

## Data availability
Data generated in this study are provided in the paper, the Supplementary Information and/or Source Data file. Source data are provided with this paper.

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

## Acknowledgements

We thank L. Schuhmacher and A. Blumhofer for fruitful discussions throughout the project. The work was supported by the Deutsche For-schungsgemeinschaft (DFG Fi459/19-1).

## Author contributions

A.L. performed most of the experiments, validated the data, wrote the first draft of the paper and supervised T. R. and K. H.. T.R. and K.H. were involved in protein purification experiments and microscopy. J.B. per-formed the histone modification experiment. K.L. co-supervised the project initially. S.E. supervised J.B. R.F. directed the project, raised the funds, supervised the co-workers, validated the data, and edited the draft of the paper.

## Funding

## Competing interests
The authors declare no competing interests.
