## [Transparent Peer Review file · Nature Communications]

The photolyase/cryptochrome of *Aspergillus nidulans* senses oxidative stress and shuttles from nuclei to mitochondria

Corresponding Author: Professor Reinhard Fischer

Version 0:

Reviewer comments:

Reviewer #1

(Remarks to the Author)

In this study, the authors investigated the roles of the fungal cryptochrome-like photolyase CryA in the model fungus *Aspergillus nidulans*. To test their role, the authors generated cryA deletion and overexpression strains and examined their development, stress responses, and gene expression profiles. Through approaches such as AlphaFold modeling and BiFC analyses, the authors demonstrated that CryA interacts with AtfA and FphA. In addition, the authors used fluorescence microscopy to observe its subcellular dynamics between the mitochondria and the nucleus. Based on these results, the authors suggest that CryA plays an important role under light or oxidative stress conditions. The findings appear to be meaningful; however, several aspects should be further addressed to strengthen the study:

- A cryA-complemented strain should be constructed and included to support the causality of the observed phenotypes.
- CryA may influence sexual development depending on environmental conditions. However, threonine medium is not optimal for observing sexual reproduction, and alternative conditions should be tested to assess CryA's role more accurately in this process.
- Although CryA interacts with FphA but not with LreA, this raises the possibility that its function may vary depending on the light wavelength. Therefore, examining both asexual and sexual development of the cryA deletion mutant under different light conditions would be valuable.
- While protein-protein interactions between CryA and FphA/AtfA were demonstrated, further genetic analyses, such as construction of double deletion mutants, are needed to clarify the genetic relationship and epistatic interactions among these components.
- In Figures 3B and 4C, it would be helpful to include corresponding DIC images to clearly show morphological details.
- In Figure 5, the N-terminal extension (NTE) of CryA appears to play a crucial role in both subcellular localization and gene expression. It is recommended to include data showing sensitivity to different concentrations of hydrogen peroxide to support this observation.
- In Figure 5A, the authors provided a magnified image of CryA-GFP. However, it would be preferable to also present magnified views of the MitoTracker and merged images to allow clearer interpretation of co-localization.
- If CryA is involved in FphA-mediated histone modification, additional supporting data should be provided to substantiate this mechanism.
- line 54, define NTE
- line 69, define CPD

Reviewer #2

(Remarks to the Author)

The manuscript by Landmark et al. describes a novel role for a CPD photolyase using the fungus *Aspergillus nidulans* as a model. Previous work had shown that this photolyase had biological functions that exceeded its original enzyme activity as a DNA repair enzyme. The phenotype of the deletion mutant showed alterations in light sensing and developmental regulation, suggesting a regulatory role for this photoreceptor. In this manuscript, the authors expand these observations to confirm that CryA is a regulator of light and stress responses in this fungus, and propose that this regulatory activity occurs, at least in part, by interacting with the red-light photoreceptor and the transcription factor AtfA. The observation of a fast transfer of CryA to mitochondria after exposure to stress is worth reporting and exploring in further detail, as the authors are surely aware. The conservation of this protein in other *Aspergilli*, and perhaps in other related fungi, suggests that this novel biological role may be conserved, at least in the relatives of *A. nidulans*. It is worth noticing the presence of a homolog gene in the human pathogen *A. fumigatus*, as indicated by the authors in figure 1. Given the relevance of the stress response in modulating the capacity of *A. fumigatus* to infect human cells, it is worth considering a continuation of this work in *A. fumigatus* which will add further relevance to the results presented in this manuscript.

The text is well-written and should be easily understood by the wide audience of the journal. The experiments are well designed and executed, and the conclusions are rightly derived from the results. I have some concerns regarding the proposed role of CryA in photoadaptation and how CryA may regulate transcription, as indicated below. Additional minor comments are also described below.

1. Results. Line 231. Photoinduction of cryA. The photoinduction of cryA shown in figure 2c is very mild, when it is compared with the genes shown in figure 2d. I suggest that the authors mention the limited effect of light (about x1.7? certainly less than two-fold).

2. Localization of CryA in nuclei: Do the authors observe any effect of light on the accumulation of cryA in nuclei? The experiments were performed in vegetative hyphae. Would the authors expect localization in mitochondria during asexual development, or after exposure to light, when hyphae are exposed to air, and oxidative stress? This observation will help to support their hypothesis about the relevance of CryA translocation during the response to oxidative stress, and provide additional support for a biological role.

3. Role of CryA as a repressor. This idea has already been proposed for CRY in *Neurospora* (Olmedo et al. *Fungal Genet Biol* 2010). If cryA acts as a repressor of FphA-dependent light regulation, as it is proposed, then the cryAOE strain should show a reduction in light-regulated expression but this is only observed in three of the genes examined and in both conditions (dark and light). Do the authors have any possible explanation? Please, discuss these observations in relation to the proposed role as a repressor.

4. The authors propose in the introduction and later in the discussion that CryA has a role in photoadaptation. This is a novel and interesting hypothesis which is supported by the interaction with both FphA and AtfA, and the increased accumulation of mRNAs for a set of light-regulated genes. However, this proposal is not further explored in the manuscript using with long-term illumination of the set of light-regulated genes in the wt, the cryA deletion or the cryAOE strain. In my opinion, the proposal that cryA participates in photoadaptation requires validation by experiments with long-term illumination where photoadaptation is observed.

5. Interaction between FphA and CryA: is this interaction transient? How do they think that this interaction regulates FphA activity? Does this interaction require light or the chromophore of either photoreceptor?

6. Interactions between AtfA and CryA. Does this interaction occur by the presence of oxidative stress? Is this interaction regulated by light? The authors show in the final section of results that CryA is shuttled to the mitochondria during exposure to oxidative stress after, presumably, losing the interaction with AtfA. Is the localization of AtfA modified in the absence of CryA?

7. Changes in the localization of CryA by oxidative stress. The authors show that the NTE and two amino acids in this segment (met and cys) participate in the changes of subcellular localization by oxidative stress. Deletion of NTE results in mitochondrial localization of CryA but mutation in the two amino acids which presumably modify the structure of the NTE results in the nuclear localization of CryA. The authors, however, only show the growth phenotype and the transcriptional profile of a set of genes for the strain with the deletion of the NTE but do not show the growth phenotype and transcriptional profile for the strain with mutations in the two amino acids of the NTE. Would the authors expect a similar phenotype and transcriptional profile for the two types of mutants in CryA?

Minor comments:

8. Introduction. Line 85: "The blue-light sensing phototropins". I suggest that they use the term "The blue-light sensing LOV proteins". I think that this is more specific.

9. Introduction. Line 85: "BLUF proteins". BLUF-domain photoreceptors have only been described in *Ustilago maydis* and a few related fungi, and their role in fungal photobiology is yet unknown. Since this is not a photoreceptor in a wide variety of fungi, I suggest that the authors indicate this in the sentence.

10. Figure 1c. There is a section of the figure in the green area that do not show the full name of the protein. It is the protein below *B. cin*. Please, correct.

11. Results. Line 175. Insert "is" at the end of the sentence: "is located in the animal..."

12. Results. Figure 2b. The legend states: "Microscopic analysis on displayed colonies on solid media (left and right) or with the Δ cryA strain grown in liquid media in the dark for 3 days (middle)." Why did the authors characterize mycelia grown in liquid media for the delta cryA strain? It would be better to show images of the three strains grown under the same conditions. Did the OE strain produce Hulle cells in liquid media?

13. Results. Figure 2d. The legend for 2d is shown as 2b. Please, correct.

14. Results. Line 245: "alteration of the protein level of CryA results in genetic mis-regulation of light- and stress-induced genes". Please, indicate the stress-induced genes that have been characterized. Figure 2c describes photoinduction in ccgA, ccgB, and conidiation genes. Is the remaining gene, AN11314 related to stress? If so, one gene does not merit indicating regulation of a whole gene category for this sentence. Later in the manuscript, figure 4, the authors show reduced mRNA accumulation for oxidative stress genes. If the authors refer to these genes, then I think that this sentence is misplaced.

15. Results. Line 279. I suggest that references 63 and 64 describing the BLI assay are placed in the previous sentence describing the method with reference 62.

16. Results. Line 303. "Recently, the expression of cryA was shown to depend on AtfA when exposed to oxidative stress". I suggest that the authors indicate the effect of this regulation: is cryA induced or repressed under these conditions?

17. Results. Figure 5e. Please, indicate the images stained with MitoTracker.

Version 1:

Reviewer comments:

Reviewer #1

(Remarks to the Author)

The authors addressed all issues raised by reviewers. The revised version can be suitable for publication in Nature Communications.

Reviewer #2

(Remarks to the Author)

The new version of the manuscript is improved compared with the initial version. All my comments and suggestions have been successfully addressed. I only have a minor comment:

Fig. 2. There are two letters "d" in the figure. The letter "d" over the photographs of plates in "dark, low O₂" should be deleted. Section "d" corresponds to the expression results by RT-PCR as shown in the text and legend.

Reviewer #1 (Remarks to the Author):

In this study, the authors investigated the roles of the fungal cryptochrome-like photolyase CryA in the model fungus *Aspergillus nidulans*. To test their role, the authors generated *cryA* deletion and overexpression strains and examined their development, stress responses, and gene expression profiles. Through approaches such as AlphaFold modeling and BiFC analyses, the authors demonstrated that CryA interacts with AtfA and FphA. In addition, the authors used fluorescence microscopy to observe its subcellular dynamics between the mitochondria and the nucleus. Based on these results, the authors suggest that CryA plays an important role under light or oxidative stress conditions. The findings appear to be meaningful; however, several aspects should be further addressed to strengthen the study:

- A *cryA*-complemented strain should be constructed and included to support the causality of the observed phenotypes.

We added the results of the re-complemented strain in Fig. 2 c, d and suppl. Fig. S2 c.

- CryA may influence sexual development depending on environmental conditions. However, threonine medium is not optimal for observing sexual reproduction, and alternative conditions should be tested to assess CryA's role more accurately in this process.

Thanks for the suggestion. The new data are shown in Fig. 2 c and suppl Fig. S2 c.

- Although CryA interacts with FphA but not with LreA, this raises the possibility that its function may vary depending on the light wavelength. Therefore, examining both asexual and sexual development of the *cryA* deletion mutant under different light conditions would be valuable.

The new data are shown in S2 c.

- While protein-protein interactions between CryA and FphA/AtfA were demonstrated, further genetic analyses, such as construction of double deletion mutants, are needed to clarify the genetic relationship and epistatic interactions among these components.

The double mutant between *cryA* and *fphA* was already included in the previous version of the paper. Please see Fig. S4 in the revised version. The *cryA/atfA* double mutant was not constructed because we already showed that in the absence of AtfA gene expression was completely abolished (Yu et al., 2019, Nat. Microbiol.).

- In Figures 3B and 4C, it would be helpful to include corresponding DIC images to clearly show morphological details.

We added the DIC pictures.

- In Figure 5, the N-terminal extension (NTE) of CryA appears to play a crucial role in both subcellular localization and gene expression. It is recommended to include data showing sensitivity to different concentrations of hydrogen peroxide to support this observation.

See new Fig. 6 c.

- In Figure 5A, the authors provided a magnified image of CryA-GFP. However, it would be preferable to also present magnified views of the MitoTracker and merged images to allow clearer interpretation of co-localization.

We added the enlarged pictures now for all images. Fig. 5 a.

- If CryA is involved in FphA-mediated histone modification, additional supporting data should be provided to substantiate this mechanism.

We did the analysis and added the data in Fig. 3 d.

- line 54, define NTE

- line 69, define CPD

Reviewer #2 (Remarks to the Author):

The manuscript by Landmark et al. describes a novel role for a CPD photolyase using the fungus *Aspergillus nidulans* as a model. Previous work had shown that this photolyase had biological functions that exceeded its original enzyme activity as a DNA repair enzyme. The phenotype of the deletion mutant showed alterations in light sensing and developmental regulation, suggesting a regulatory role for this photoreceptor. In this manuscript, the authors expand these observations to confirm that CryA is a regulator of light and stress responses in this fungus, and propose that this regulatory activity occurs, at least in part, by interacting with the red-light photoreceptor and the transcription factor AtfA.

The observation of a fast transfer of CryA to mitochondria after exposure to stress is worth reporting and exploring in further detail, as the authors are surely aware. The conservation of this protein in other *Aspergilli*, and perhaps in other related fungi, suggests that this novel biological role may be conserved, at least in the relatives of *A. nidulans*.

It is worth noticing the presence of a homolog gene in the human pathogen *A. fumigatus*, as indicated by the authors in figure 1. Given the relevance of the stress response in modulating the capacity of *A. fumigatus* to infect human cells, it is worth considering a continuation of this work in *A. fumigatus* which will add further relevance to the results presented in this manuscript.

Thanks for this suggestion.

The text is well-written and should be easily understood by the wide audience of the journal. The experiments are well designed and executed, and the conclusions are rightly derived from the results. I have some concerns regarding the proposed role of CryA in photoadaptation and how CryA may regulate transcription, as indicated below. Additional minor comments are also described below.

1. Results. Line 231. Photoinduction of cryA. The photoinduction of cryA shown in figure 2c is very

mild, when it is compared with the genes shown in figure 2d. I suggest that the authors mention the limited effect of light (about x1.7? certainly less than two-fold).

We added that aspect.

2. Localization of CryA in nuclei: Do the authors observe any effect of light on the accumulation of cryA in nuclei? The experiments were performed in vegetative hyphae. Would the authors expect localization in mitochondria during asexual development, or after exposure to light, when hyphae are exposed to air, and oxidative stress? This observation will help to support their hypothesis about the relevance of CryA translocation during the response to oxidative stress, and provide additional support for a biological role.

Thanks for this idea. We analyzed the localization of CryA in different light conditions and found the protein always in nuclei. We also investigated the localization in conidiophores and also found the protein only in nuclei. See new Fig. S2 a.

3. Role of CryA as a repressor. This idea has already been proposed for CRY in *Neurospora* (Olmedo et al. *Fungal Genet Biol* 2010). If cryA acts as a repressor of FphA-dependent light regulation, as it is proposed, then the cryAOE strain should show a reduction in light-regulated expression but this is only observed in three of the genes examined and in both conditions (dark and light). Do the authors have any possible explanation? Please, discuss these observations in relation to the proposed role as a repressor.

We repeated this set of experiments and used less conidia as inoculum and kept the conditions like in all other qRT PCR experiments because we added the re-complemented strain. Under these conditions the overexpression strain indeed shows reduced expression of the studied genes. In general gene expression was lower than in the previous experiment.

4. The authors propose in the introduction and later in the discussion that CryA has a role in photoadaptation. This is a novel and interesting hypothesis which is supported by the interaction with both FphA and AtfA, and the increased accumulation of mRNAs for a set of light-regulated genes. However, this proposal is not further explored in the manuscript using with long-term illumination of the set of light-regulated genes in the wt, the cryA deletion or the cryAOE strain. In my opinion, the proposal that cryA participates in photoadaptation requires validation by experiments with long-term illumination where photoadaptation is observed.

Thank you very much for this suggestion. We repeated the experiment and quantified the transcripts also after longer light exposure and found that also in the absence of CryA expression of *ccgA* and *brlA* only affects the early time point but not the later ones. This suggests a repressing function of the light and development induction but not really a long-term adaptation.

5. Interaction between FphA and CryA: is this interaction transient? How do they think that this interaction regulates FphA activity? Does this interaction require light or the chromophore of either photoreceptor?

Our BLI experiments show that the interaction *in vitro* is reversible, but we have no evidence for that *in vivo* because the split YFP system does not allow to answer that question. We tested whether the interaction was influenced by light or the presence of the chromophore in FphA. The results were

negative *in vivo* (mentioned in the text), and the interaction strength was largely reduced *in vitro* (Fig. S4 b).

6. Interactions between AtfA and CryA. Does this interaction occur by the presence of oxidative stress? Is this interaction regulated by light? The authors show in the final section of results that CryA is shuttled to the mitochondria during exposure to oxidative stress after, presumably, losing the interaction with AftA. Is the localization of AftA modified in the absence of CryA?

The localization of AtfA was not changed in the absence of CryA (Fig. S5 a). Light did not have any effect on the interaction. The addition of H₂O₂ caused the disappearance of the nuclear signal. However, YFP is very sensitive towards hydrogen peroxide and hence this experiment is not very meaningful.

7. Changes in the localization of CryA by oxidative stress. The authors show that the NTE and two amino acids in this segment (met and cys) participate in the changes of subcellular localization by oxidative stress. Deletion of NTE results in mitochondrial localization of CryA but mutation in the two amino acids which presumably modify the structure of the NTE results in the nuclear localization of CryA. The authors, however, only show the growth phenotype and the transcriptional profile of a set of genes for the strain with the deletion of the NTE but do not show the growth phenotype and transcriptional profile for the strain with mutations in the two amino acids of the NTE. Would the authors expect a similar phenotype and transcriptional profile for the two types of mutants in CryA?

Thanks for this aspect. Meanwhile we could further specify the effect because we found that cysteine and not methionine plays the crucial role in the regulation (Fig. 6 b). We checked the phenotypes on agar plates (Fig. 6 c) and analyzed the expression of the ROS-responding genes. This revealed many new interesting insights into the function of the NTE and the cysteine.

Minor comments:

8. Introduction. Line 85: “The blue-light sensing phototropins”. I suggest that they use the term “The blue-light sensing LOV proteins”. I think that this is more specific.

done

9. Introduction. Line 85: “BLUF proteins”. BLUF-domain photoreceptors have only been described in *Ustilago maydis* and a few related fungi, and their role in fungal photobiology is yet unknown. Since this is not a photoreceptor in a wide variety of fungi, I suggest that the authors indicate this in the sentence.

done

10. Figure 1c. There is a section of the figure in the green area that do not show the full name of the protein. It is the protein below *B. cin.* Please, correct.

done

11. Results. Line 175. Insert “is” at the end of the sentence: “is located in the animal...”

done

12. Results. Figure 2b. The legend states: “Microscopic analysis on displayed colonies on solid media (left and right) or with the $\Delta cryA$ strain grown in liquid media in the dark for 3 days (middle).” Why did the authors characterized mycelia grown in liquid media for the delta cryA strain? It would be better to show images of the three strains grown under the same conditions. Did the OE strain produce Hulle cells in liquid media?

We now compare the phenotypes on agar plates (Fig. 2 c). Just for information, the OE strain does not produce Huelle cells in liquid media.

13. Results. Figure 2d. The legend for 2d is shown as 2b. Please, correct.

corrected

14. Results. Line 245: “alteration of the protein level of CryA results in genetic mis-regulation of light- and stress-induced genes”. Please, indicate the stress-induced genes that have been characterized. Figure 2c describes photoinduction in *crgA*, *crgB*, and conidiation genes. Is the remaining gene, AN11314 related to stress? If so, one gene does not merit indicating regulation of a whole gene category for this sentence. Later in the manuscript, figure 4, the authors show reduced mRNA accumulation for oxidative stress genes. If the authors refer to these genes, then I think that this sentence is misplaced.

AN11314 is another light-induced gene. It is now explained in the text.

15. Results. Line 279. I suggest that references 63 and 64 describing the BLI assay are placed in the previous sentence describing the method with reference 62.

done

16. Results. Line 303. “Recently, the expression of *cryA* was shown to depend on *AtfA* when exposed to oxidative stress”. I suggest that the authors indicate the effect of this regulation: is *cryA* induced or repressed under these conditions?

done

17. Results. Figure 5e. Please, indicate the images stained with MitoTracker.

done

Reviewer #1 (Remarks to the Author):

The authors addressed all issues raised by reviewers. The revised version can be suitable for publication in Nature Communications.

thank you

Reviewer #2 (Remarks to the Author):

The new version of the manuscript is improved compared with the initial version. All my comments and suggestions have been successfully addressed. I only have a minor comment:

Fig. 2. There are two letters "d" in the figure. The letter "d" over the photographs of plates in "dark, low O₂" should be deleted. Section "d" corresponds to the expression results by RT-PCR as shown in the text and legend.

We deleted the "d".